# Keratin 8/18a.1 Expression Influences Embryonic Neural Crest Cell Dynamics and Contributes to Postnatal Corneal Regeneration in Zebrafish

**DOI:** 10.3390/cells13171473

**Published:** 2024-09-02

**Authors:** Antionette L. Williams, Brenda L. Bohnsack

**Affiliations:** 1Division of Ophthalmology, Ann & Robert H. Lurie Children’s Hospital of Chicago, 225 E. Chicago Ave., Chicago, IL 60611, USA; bbohnsack@luriechildrens.org; 2Department of Ophthalmology, Feinberg School of Medicine, Northwestern University, 645 N. Michigan Ave., Chicago, IL 60611, USA

**Keywords:** cytokeratin 8, cytokeratin 18a.1, keratin intermediate filaments, neural crest cells, eye development, anterior segment, cornea, ophthalmic injury repair, regenerative medicine, zebrafish

## Abstract

A complete understanding of neural crest cell mechanodynamics during ocular development will provide insight into postnatal neural crest cell contributions to ophthalmic abnormalities in adult tissues and inform regenerative strategies toward injury repair. Herein, single-cell RNA sequencing in zebrafish during early eye development revealed keratin intermediate filament genes *krt8* and *krt18a.1* as additional factors expressed during anterior segment development. In situ hybridization and immunofluorescence microscopy confirmed *krt8* and *krt18a.1* expression in the early neural plate border and migrating cranial neural crest cells. Morpholino oligonucleotide (MO)-mediated knockdown of K8 and K18a.1 markedly disrupted the migration of neural crest cell subpopulations and decreased neural crest cell marker gene expression in the craniofacial region and eye at 48 h postfertilization (hpf), resulting in severe phenotypic defects reminiscent of neurocristopathies. Interestingly, the expression of K18a.1, but not K8, is regulated by retinoic acid (RA) during early-stage development. Further, both keratin proteins were detected during postnatal corneal regeneration in adult zebrafish. Altogether, we demonstrated that both K8 and K18a.1 contribute to the early development and postnatal repair of neural crest cell-derived ocular tissues.

## 1. Introduction

A critical barrier in developing vision-saving strategies for ophthalmic injury repair is the limited knowledge of the cellular changes underlying the destruction of normal cellular architecture and replacement with scar tissue leading to fibrosis. In the eye, fibrosis underlies the pathogenesis of many blinding conditions, including corneal and conjunctival scarring, Fuchs’ endothelial dystrophy, primary open-angle glaucoma, age-related macular degeneration, and proliferative vitreoretinopathy [1,2,3,4]. Notably, many features of ophthalmic wound healing recapitulate those of ocular development [5]. During early embryogenesis, cranial neural crest cells within and surrounding the developing eye contribute to the formation of ocular anterior segment tissues, including the cornea, iris, ciliary body and muscles, trabecular meshwork, and sclera [6,7,8,9]. Accumulating evidence has revealed that the embryonic origins of tissues significantly affect their postnatal potential for regeneration and repair [10,11]. Thus, with respect to the eye, the ability to replace its tissues likely relates to the contributions of postnatal resident neural crest cells, as a pluripotent embryonic remnant cell population with the capacity for self-renewal and differentiation along specified molecular pathways lifelong [8,9,12,13,14,15]. However, despite increasing evidence that the stem cell-like features of neural crest cells persist into adulthood [14,16,17], many of the essential players that afford the initial and continued plasticity of these cells remain elusive. Thus, a complete understanding of neural crest cell signaling and mechanodynamics during ocular development will provide insight into their postnatal activities during ophthalmic injury repair. To determine the feasibility and viability of this idea, here we employed single-cell RNA sequencing technology and embryonic zebrafish to identify novel additional factors regulating neural crest contributions to eye development. From our analysis, we identified keratin intermediate filament genes, particularly *krt8* and *krt18*, as players in ocular neural crest development and the formation of the anterior segment. We further highlighted postnatal roles for these genes in the injury repair of neural crest-derived corneal tissues in adult zebrafish eyes. Taken together, our study provides the first evidence for the involvement of *krt8* and *krt18a.1* in neural crest cell dynamics during anterior segment development and postnatal corneal regeneration in the zebrafish.

## 2. Materials and Methods

### 2.1. Animal Husbandry and Strain Description

These studies utilize zebrafish (*Danio rerio*) maintained in our aquatics facility at 28.5 degrees Celsius on a 14/10 h light/dark cycle for suitable growth and breeding. Embryos collected from spawning were maintained in an incubator at 28.5 degrees Celsius and subsequently morphologically staged using hours postfertilization (hpf) [18]. For analyses involving in situ hybridization and live-cell fluorescence imaging, the embryos were dechorionated and subsequently treated with 0.003% phenylthiourea solution at 22 hpf to inhibit pigmentation in the eye [19]. The following strains Tg(-*4.7sox10*::EGFP), Tg(*foxd3*::GFP), and Tg(*rx3*::GFP) were, respectively, gifted from Dr. Mary Halloran (PhD, University of Wisconsin, Madison, WI, USA), Dr. Thomas Schilling (PhD, University of California, Irvine, CA, USA), and Dr. Steve Wilson (PhD, University College London, United Kingdom). To reduce native fluorescence and natural pigmentation interference, these strains were crossed into the *roy*-/-, *nacre*-/- (Casper) double homozygous pigmentation mutant background [20,21], a strain which in all other aspects is phenotypically wildtype. All animal study protocols were previously approved by the Institutional Animal Care and Use Committee (IACUC, Protocol #IS00015598) of Northwestern University, an AALAC-accredited institution, in affiliation with Stanley Manne Children’s Research Institute (Chicago, IL, USA). The experiments described herein were performed according to the Public Health Service Policy on Humane Care and Use of Laboratory Animals and the Guide for the Care and Use of Laboratory Animals.

### 2.2. Transcriptome Analysis

#### 2.2.1. Fluorescence-Activated Cell Sorting

The zebrafish embryos from strains Tg(-*4.7sox10*::EGFP) and Tg(*foxd3*::GFP) were harvested at 48 hpf, decapitated, and the heads were collected and dissociated into single cells using FACSmax^TM^ cell dissociation solution (AMSBIO, Cambridge, MA, USA). Fluorescence-activated cell sorting (FACS) was performed on a BD FACSAria SORP system and BD FACSymphony S6 SORP system and purchased through the support of NIH 1S10OD011996-01 and 1S10OD026814-01. Green fluorescent protein (GFP)-positive cells were obtained with gates defined by cells from negative control wildtype Casper strain embryos. To obtain an adequate number of cells per sample, the heads from ~500 embryos per transgenic line were collected.

#### 2.2.2. Library Preparation and Sequencing

Samples for single-cell RNA sequencing (scRNAseq) were examined for cell viability, density, and prep quality (i.e., no clumps, cell debris, etc.). Samples passing the initial quality check were loaded onto the 10× Genomics Chromium platform (10× Genomics, Pleasanton, CA, USA) for the partitioning and encapsulation of single cells into nanoliter-sized GEMs (Gel beads-in-Emulsion; 10× Genomics). Each encapsulated cell was lysed within its GEM, and total RNA was released and then reverse transcribed to cDNA using primers attached to gel beads carrying a unique 10× barcode for downstream cell separation. Subsequently, the GEMs were broken, and all uniquely barcoded cDNAs were pooled. The cDNA pools were PCR amplified to generate enough material for Illumina sequencing. The sequencing library construction process was carried out using the 10× Genomics Single Cell 3’ v3.1 Protocol. Sequencing of the 10× libraries was performed in the Northwestern University Sequencing Core (NUSeq, Feinberg School of Medicine, Center for Genetic Medicine, Chicago, IL, USA) at a depth of ~30,000 reads per cell.

#### 2.2.3. Single-Cell RNA Sequencing Analysis 

Raw sequencing data, in base call format (.bcl) was demultiplexed using Cell Ranger (10× Genomics), converting the raw data into the FASTQ format. Cell Ranger was also used to align the FASTQ files to the zebrafish reference genome (danRer11) and count the number of reads from each cell that aligned to each gene. The Loupe Browser (version 7.0.1; 10× Genomics) was used to evaluate the data and for further downstream analysis. Pathway enrichment analysis was performed using the free gene annotation and analysis resource Metascape [22]. Protein–protein association and functional enrichment analyses were performed using the free STRING database [23], which covers approximately 68 million proteins from over 14,000 organisms.

### 2.3. Imaging and Image Processing

Wholemount imaging and phenotypic analysis were performed with an automated fluorescence stereomicroscope (M205FA, Leica Microsystems CMS GmbH, Wetzler, Germany) equipped with DFC290 (Leica) ORCA-ER (Hamamatsu, Hamamatsu City, Japan) cameras to obtain brightfield and fluorescent images, respectively. Thin-section analysis was performed using a confocal laser scanning microscope (LSM 880 Carl Zeiss AG, Oberkochen, Germany). Z-stacks obtained from the corneal lateral edge to 100 μm medial to the ocular edge were deconvolved and, subsequently, max projected for the acquisition of a single image. ZEN (Black Edition, v2.0) imaging software (Carl Zeiss AG) and Photoshop 2021 editing and design software (Adobe Systems Incorporated, San Jose, CA, USA) were utilized for image processing and quantitative analyses. The images are shown as representatives of the results obtained from all experiments. Images from at least 4–6 embryos/group were included in immunofluorescence quantifications. Statistical analyses were performed using Microsoft Excel for Mac (v.16.8; Microsoft Corporation, Redmond, WA, USA), considering *p* < 0.05 as statistically significant.

### 2.4. Wholemount In Situ Hybridization and Immunostaining

For wholemount in situ hybridization experiments, digoxigenin (DIG)-labeled (Roche Life Science, MilliporeSigma, Burlington, MA, USA) antisense RNA probes targeting zebrafish *krt8* (NM_200080.2; Forward: 5′-ACAAGGCAGGATGTCCACCTACAGC-3′, Reverse: 5′-TTGCATGTTGTGCAGGTCAGCCTCCAGC-3′), *krt18a.1* (NM_178437.2; Forward: 5′-ACCAGCGACCATGAGTCTGAGAACAAGC-3′, Reverse: 5′-TGACCTTTCTTAGGCCGGTGATGTCGGC-3′), or *foxd3* (BC_095603.1; Forward: 5′-GAGATCGAGGTGAAGGAG-3′, Reverse: 5′-TAATACGACTCACTATAGGGTGTGGACGCTGTCGGTAAAA-3′) were prepared by in vitro transcription using T7 polymerase (Roche Life Science). Staged, paraformaldehyde (PFA)-fixed (4% in 1X phosphate-buffered saline, PBS; overnight at 4 degrees Celsius) zebrafish embryos (Casper, Tg(-*4.7sox10*::EGFP), Tg(*foxd3*::EGFP) and Tg(*rx3*::GFP)) were subsequently washed in 1X PBS and passed through a methanol series (25%, 50%, 75%, and 100% in 1X PBS, 5 min, nutating at room temperature for each step). The last step was replaced with fresh 100% methanol, and then the embryos were stored at -20 degrees Celsius for at least overnight. For the experiment, the stored embryos were rehydrated through 75% methanol:25% 1X PBS, 50% methanol:50% 1X PBS, and 25% methanol:75% 1X PBS solutions for 5 min, nutating at room temperature, followed by a final incubation in 100% PBS and washing in 1X PBS containing 0.1% Tween 20 (PBST; 3 × 5 min, nutating at room temperature). To permeabilize the tissues and allow access to the RNA probe, the embryos were subsequently digested with 10 μg/mL proteinase K (RNA-grade solution; ThermoFisher Scientific, Waltham, MA, USA) nutating at room temperature for varying digestion times according to the developmental stage. In our hands, for embryos up to 24 hpf, digestion is not necessary, whereas for embryos 36 hpf, a 30 s digestion is sufficient, and for embryos 48 hpf, a 1 min digestion is sufficient. Digestion, when performed, was stopped by washing with 1X PBST, followed immediately by the refixation of all embryos in 4% PFA, nutating at room temperature for 20 min. The fixed embryos were then washed with 1X PBST, 1× for 10 min, nutating at room temperature, and transferred to fresh 1.5 mL Eppendorf Safe-Lock microcentrifuge tubes (USA Scientific, Inc., Ocala, FL, USA) at 50 embryos/tube per experimental condition. Prehybridization was performed by incubation in 500 μL hybridization buffer in a hybridization oven at 56 degrees Celsius with end-over-end rotation for at least 30 min. The prehybridization buffer is removed and replaced with 250 μL hybridization buffer containing approximately 100 ng of antisense RNA probe and incubated overnight at 56 degrees Celsius with end-over-end rotation. The next day, the RNA probe was removed, and the embryos were subjected to the following stringency washes in a hybridization oven at the indicated temperatures with end-over-end rotation: (1) 50% 2X saline sodium citrate (SSC) buffer:50% formamide at 65 degrees Celsius for 1 h; (2) 2X SSC at 37 degrees Celsius for 3 × 10 min each; (3) RNase digestion (10 μg/mL in RNase dilution buffer containing 0.1% Tween 20, RNase T) at 37 degrees Celsius for 1 h; (4) RNase T buffer alone at 65 degrees Celsius for 30 min; and (5) 2X SSC at 37 degrees for 10 min. Final washes in PBST were performed with nutating at room temperature for 3 × 5 min each. The embryos were then transferred to 1.5 mL siliconized microcentrifuge tubes and blocked with 500 μL Maleate/0.1% Triton/Blocking reagent (MilliporeSigma) for 2 h, nutating at room temperature. The blocking buffer was subsequently removed, and the embryos were incubated with 250 μL with anti-digoxigenin antibody conjugated to alkaline phosphatase (1:1000 in blocking buffer; cat. no. 11-093-274-910; Sigma-Aldrich, Inc., St. Louis, MO, USA). After removing the antibody solution and extensively washing with 1X Maleic Acid Buffer containing 0.1% Triton (MABT), alkaline phosphatase activity was evaluated using the Vector Blue (SK-5300) or Vector Red (SK-5100) Alkaline Phosphatase Substrate Kit (Vector Laboratories, Inc., Newark, CA, USA), which is both chromogenic (blue or red, respectively) and fluorescent (Cy5 or Texas Red, respectively). For chromogenic reactions, the embryos were developed for equal amounts of time. Sense RNA probe control experiments were also performed and developed in parallel to ensure specific antisense RNA probe binding. Each experiment was conducted a minimum of 3 times.

Where indicated, immunohistochemistry was performed on whole embryos after in situ hybridization analyses. Briefly, the embryos were washed in PBST (3 × 5 min) at room temperature, fixed in 4% PFA for 30 min, and washed again with 1X PBST for 10 min at room temperature, followed by blocking in 1X PBST blocking solution containing 10% goat serum (Invitrogen, Carlsbad, CA, USA), and 0.5% BSA (bovine serum albumin; Sigma-Aldrich) for at least 2 h, nutating at room temperature. After blocking, the embryos were incubated in a rabbit polyclonal antibody against GFP (1:100 in 1X PBST blocking solution; ab6556; Abcam, Cambridge, UK) with end-over-end rotation overnight at 4 degrees Celsius. The next day, the embryos were washed with 1X PBST (3 × 10 min) nutating at room temperature, and then incubated with goat anti-rabbit IgG (H + L) cross-adsorbed secondary antibody Alexa Fluor™ 488 (1:200 in 1X PBST blocking solution; Life Technologies, Carlsbad, CA, USA) for at least 2 h nutating at room temperature. After washing with 1X PBST (3 × 10 min) nutating at room temperature, the embryos were cryoprotected in successive sucrose solutions (5% and 20% in 1X PBS), embedded in Tissue Tek^®^ Optimal Cutting Temperature (O.C.T) compound (Sakura Finetek USA, Inc., Torrance, CA, USA)), and then sectioned at 10 μm rostrocaudally and perpendicular to the spine to obtain transverse sections through the head. The sections were covered with mounting media (ProLong™ Gold Antifade Mountant with DNA stain DAPI (40,6-diamidine-2-phenylidole-dihydrochloride) ThermoFisher Scientific), coverslipped, and subsequently imaged according to the description above.

### 2.5. Pharmacological Treatments

For experiments involving pharmacological treatments, all-trans retinoic acid (ATRA; R2625; Sigma-Aldrich) and the reversible aldehyde dehydrogenase inhibitor 4-diethylaminobenzaldehyde (DEAB; D86256; Sigma-Aldrich) were prepared as 1000x stock solutions in dimethyl sulfoxide (DMSO; Sigma-Aldrich). For each pharmacologic agent, the concentrations used in the final experiments (i.e., 100 nM ATRA and 10 μM DEAB) are predicated on the LD50 and phenotype consistency and are consistent with the concentrations used in previous experiments resulting in the same phenotype [24]. The final concentration of 0.1% DMSO was used as a vehicle control. All pharmacological treatments were conducted in dark conditions because of the light sensitivity of the reagents. For embryonic experiments, Casper embryos, at 50 to 100 embryos per treatment group, were dechorionated, and pharmacologically treated between 24 and 27 hpf (as described in Section 3. Results). The experiments were repeated 4–6 times. At 48 hpf, the phenotypes were assessed by wholemount in situ hybridization, followed by cryosectioning and imaging as described above. For experiments involving adult zebrafish, an equal number of male and female, age-matched (> 90 days postfertilization) fish were treated by immersion in a minimum of 250 mL of freshly prepared treatment solution (in system water). Experiments used 4–6 adult fish per treatment group and were repeated a minimum of 3 times. The fish were euthanized after 24 h of treatment. The eyes were harvested, fixed in 4% PFA overnight at 4 degrees Celsius, subsequently cryoprotected in successive sucrose solutions, embedded in O.C.T compound, and then sectioned in an orientation with the lens facing outward to obtain transverse sections at 10 μm. The sections were mounted with mounting media containing DAPI, coverslipped, and imaged as described above.

### 2.6. Morpholino Oligonucleotide Injections

Translation-blocking antisense morpholino oligonucleotides (MOs) targeting the promoter regions of zebrafish *krt8* (5′-TTTTCTTGCTGTAGGTGGACATCCT-3′) and *krt18a.1* (5′-GTAGCTTGTTCTCAGACTCATGGTC-3′) and a lissaminated antisense control MO (hemoglobin; 5′-CCTCTTACCTCAgTTACAATTTATA-3′) (Gene Tools, LLC, Philomath, OR, USA) were reconstituted to 1 mM stock solution in distilled water. For each MO, working concentrations generating consistent, reproducible phenotypes were resolved, and 1 to 2 nL at 0.25 mM (2.1 ng/nL) was injected into Casper Tg(-*4.7sox10*::EGFP), Tg(*foxd3*::GFP), and Tg(*rx3*::GFP) embryos at the single cell stage. To detect altered phenotypes, the embryos were imaged at 48 hpf as described above. Each experiment was conducted a minimum of 3 times with at least 25 to 50 embryos per group.

### 2.7. Corneal Abrasion and Immunohistochemistry

Mechanical corneal injury [25,26] was performed using adult zebrafish. Briefly, the fish were sedated by immersion in 0.02% MS-222 (Tricaine; Syndel, Ferndale, WA, USA). Corneal abrasion was performed using an Algerbrush II ocular burr (0.5 mm; Precision Vision, Woodstock, IL, USA). Corneal fluorescein staining (BioGlo Fluorescein Sodium Ophthalmic Strips; 1 mg; Sigma Pharmaceuticals, North Liberty, IA, USA) was used to check for any disruptions on the corneal surface. The fish were recovered in system water containing lidocaine hydrochloride monohydrate (2.5 mg/L; Sigma-Aldrich) and euthanized at the indicated times postinjury. Each experiment was repeated 3–4 times, harvesting 4–6 fish at each collection period. The eyes were subsequently harvested, fixed by overnight incubation in 4% PFA with end-over-end rotation at 4 degrees Celsius, in successive sucrose solutions (5% and 20% in 1X PBS), embedded in Tissue Tek^®^ O.C.T compound, and then sectioned at 10 μm in an orientation with the lens facing outward to obtain transverse sections. To examine keratin expression during wound healing, the sections were subjected to immunostaining with a monoclonal (mouse) anti-K8 antibody (1:100; GTX34663; GeneTex, Irvine, CA, USA) and a polyclonal (rabbit) anti-K18a.1 antibody (1:500; GTX112978; GeneTex), both with reactivity to zebrafish, according to in-house specificity analysis (anti-K8; Appendix A) and the manufacturer’s specifications (anti-K18; Appendix A). The sections were fixed by incubation in ice-cold acetone (10 min on ice), followed by washing with 1X PBS (2 × 5 min at room temperature) and blocking in 1X PBS containing 5% goat serum and 0.3% Triton at room temperature for 1 h. Subsequently, the blocking solution was removed, and primary antibody incubation was performed overnight at 4 degrees Celsius. The next day, the sections were washed with 1X PBS (3 × 5 min at room temperature) and then incubated with goat anti-mouse or rabbit IgG (H + L) cross-adsorbed secondary antibody Alexa Fluor™ 488 or 647 (1:500 in blocking solution; Life Technologies) at room temperature for 2 h. After washing for 3 × 5 min with 1X PBS at room temperature, the sections were covered with mounting media (ProLong™ Gold Antifade Mountant with DNA stain DAPI (40,6-diamidine-2-phenylidole-dihydrochloride) ThermoFisher Scientific), coverslipped, and subsequently imaged according to the description above. 

## 3. Results

### 3.1. Single Cell Data Atlas Reveals the Differential Expression of Keratin Intermediate Filament Genes during Ocular Neural Crest Development

To examine transcriptional changes taking place during ocular anterior segment morphogenesis and identify the involvement of novel factors, we isolated and analyzed the single-cell gene expression profiles of migrating neural crest subpopulations in the head mesenchyme during early embryogenesis. To precisely sample these cells, we utilized transgenic zebrafish reporter lines expressing GFP under the transcriptional control of *sox10* (SRY box 10) and *foxd3* (Forkhead box d3), marker genes that have previously been shown to regulate early ocular neural crest differentiation [7,8,9,27,28,29,30,31,32,33,34,35]. Live images taken at 48 hpf, a developmental timepoint at which anterior segment subdifferentiation/lineage diversification initiates, showed that *sox10* and *foxd3* are highly expressed in progenitor neural crest cells in the craniofacial and periocular/ocular regions of developing zebrafish embryos (*inset*, Figure 1A), confirming that both transcription factors are adequate representatives of the periocular mesenchyme (POM) responsible for anterior segment formation. Therefore, we decapitated 48-hpf Tg(*foxd3*::GFP) and Tg(-*4.7sox10*::EGFP) embryos and dissociated the tissues into single cells. *Sox10*- and *foxd3*-GFP+ neural crest cells were subsequently isolated by FACS, and the purified GFP+ neural crest cells were processed using the 10× Genomics Chromium platform to generate scRNA libraries (Figure 1A). Three biological replicates for each sample (~3000 embryos total) were independently collected. Sequencing results showed ~8500 cells captured from the *foxd3*+ sample while 10,750 cells were captured from the *sox10*+ sample, with a count of 19,250 cells for the total data set. Approximately 5000+ genes were identified from ~30,000+ reads/cell for each sample. The sequencing data was further processed using Cell Ranger (10× Genomics), and UMAP-based cluster distribution was analyzed using Loupe Browser. 

Notably, as *sox10* and *foxd3* represent different neural crest cell subpopulations within the POM, we combined the two data sets for cluster distribution analysis to evaluate the two populations together and determine whether they remain discrete or cluster together. The results showed that both *sox10 (light blue)* and *foxd3* (*magenta*) neural crest cell subpopulations primarily clustered together (*purple*), with few non-overlapping sox10+ and foxd3+ clusters (left panel, Figure 1B). Further analysis revealed 21 gene clusters, representing functional subgroups among these periocular/ocular neural crest cell subpopulations (*right*, Figure 1B). Cell type analysis was based on the upregulated expression of a collection of other POM transcription factors (*pom+*; left panel, Figure 1C), namely, *pitx2* (paired-like homeodomain 2), *foxc1* (forkhead box c1), *lmx1b* (LIM homeobox transcription factor 1 beta), and *eya2* (EYA transcriptional coactivator and phosphatase 2), previously implicated in neural crest cell signaling during early ocular development (reviewed in [7,8,9,34,36]). This analysis revealed 7 clusters of interest (COIs), showing at least 2.5-fold upregulated median expression of the other POM genes (*y-axis*, right panel, Figure 1C) (*bolded font*, *x-axis*, right panel, Figure 1C). Pathway enrichment analysis (Metascape) revealed gene signatures associated with events leading up to and occurring during ocular anterior segment morphogenesis, namely, ‘optic fissure closure’ (Clusters 2, 7), ‘pharyngeal development’ (Clusters 3, 10), ‘mesoderm development’ (Cluster 7), ‘neural crest development/endothelial cell proliferation’ (Cluster 12), ‘eye morphogenesis/endothelial migration’ (Cluster 16), and ‘cell migration’ (Clusters 2 and 10) (*black font*, Figure 1D). Signaling through RA, transforming growth factor-beta (TGFβ) and Wnt was also associated with all functional clusters, consistent with control of the positional identity of neural crest cells during eye development. Additionally, the expression of genes related to physiological processes, including ‘collagen organization’ (Cluster 10) and ‘extracellular matrix remodeling’ (Clusters 2, 3, 7, 9, 12, and 16), which are important for development, and normal organ homeostasis were also identified (*gray font*, Figure 1D). While associations with ‘stem cell/cell migration’, ‘regeneration’, and ‘wound response’ further supported the participation of neural crest cells in the healing and repair of ocular tissues (*blue font*, Figure 1D), as previously suggested [11]. Most notably, ‘intermediate filament-based processes’ were associated with ‘eye morphogenesis’, ‘ECM remodeling’, and ‘regeneration’ (Cluster 9) (*magenta font*, Figure 1D). Further analysis of functional protein associations (STRING) revealed cytokeratin intermediate filament gene networks comprising *krt94*, *krt8*, *krt18a.1,* and *krt18b* in various combinations among the clusters associated with ‘eye morphogenesis’ and ‘neural crest development’, namely, Clusters 9, 12, and 16 (Figure 1E), with *krt8*, *krt18a.1*, and *krt18b* showing the highest expression levels (Figure 1F). 

Keratins are the fibrous protein-forming units of intermediate filaments, one of three well-characterized cytoskeletal networks that enable cells to withstand deformation [37,38,39,40,41]. Reference to the Zebrafish Information Network (ZFIN), a database of reported and predicted genetic and genomic information for the model organism zebrafish (Danio rerio), showed that *krt8* and *krt18a.1* expression had previously been detected in the eyes of adult zebrafish at > 90 days postfertilization (dpf) [42], whereas *krt18b* expression was previously detected throughout the entire juvenile zebrafish at 30–45 dpf but not in the eyes. Nonetheless, these data identified keratin intermediate filament genes as potential constituents of the POM milieu.

However, no roles for keratin intermediate filament genes in ocular anterior segment formation have been reported, and few studies have identified the players responsible for the structural integrity of cranial and ocular neural crest cells. Moreover, the most widely dispersed members of the intermediate filament gene family *krt8* and *krt18*, although previously reported to be expressed in the zebrafish eye, have not been demonstrated to play a role in neural crest development or ocular anterior segment morphogenesis. Notably, we only examined the potential roles of *krt8* and *krt18a.1* in the ocular neural crest since *krt18b* expression in the zebrafish eye has not been previously reported. Furthermore, both *krt18a.1* and *18b* are human KRT18 orthologs and transcript variants of the same protein that were equally upregulated in the POM neural crest during ocular development (Figure 1F).

### 3.2. Krt8 and krt18a.1 Are Expressed at the Early Neural Plate Border and in Migrating Neural Crest Cells during Early Development in Zebrafish

To determine roles for *krt8* and *krt18a.1* in the neural crest during anterior segment development, we first examined their gene expression patterns in zebrafish embryos at 8, 12, 16, 20, 24, and 48 hpf, time points corresponding to early ocular development (~1200 embryos total). Wholemount images of chromogenic (brightfield, blue) in situ hybridization showed *krt8* and *krt18a.1* expression on the dorsal side of the zebrafish embryo along the neural plate border and enveloping layer (EVL) at 8 hpf (Figure 2A,G) and then on the dorsoposterior side in the embryonic epithelium at 20–24 hpf, with marked krt18a.1 expression in the lateral line. Ventral ocular and craniofacial *krt8* and *krt18a.1* expression (Figure 2B–F,H–L) with apparent expression in the anterior segment and facial mesenchyme (*blue arrows*, Figure 2F,L) was detected in the developing zebrafish by 48 hpf. Section fluorescent double in situ hybridization at 24 hpf for *krt8* or *krt18a.1* (Cy5; *dark blue*) and *foxd3* (Texas Red; *magenta*) expression, followed by immunohistochemistry for *sox10*-GFP (*light blue*), showed *krt8* and *krt18a.1* expression in the primordial cornea (*orange arrows*, Figure 2M,N) attached to the surface of the emerging spherical lens and among *foxd3*- and *sox10*-positive neural crest cell subpopulations in the periocular region (*yellow arrows*, Figure 2M,N). Thus, in addition to their well-characterized epithelial localization, *krt8* and *krt18a.1* were also expressed in the ocular and periocular neural crest during early anterior segment development in zebrafish.

### 3.3. MO Knockdown of K8 and K8a.1 Disrupts Neural Crest Migration in the Ocular and Craniofacial Regions during Early Development

We next employed a transient knockdown approach to examine the function of these keratin intermediate filament proteins in the neural crest during early ocular development in Casper zebrafish. To knockdown K8 or K18a.1 expression, antisense MOs targeting the promoter regions of *krt8* and *krt18a.1* were separately injected into Casper zebrafish embryos at the single-cell stage (~600 embryos total). Examination of the phenotypic effects of K8 and K18a.1 MO knockdown at 48 hpf revealed that compared with uninjected and control-injected embryos (Figure 3A,B), MO-injected embryos exhibited coloboma (*arrowhead*, Figure 3C) and unusually small (*asterisk*, Figure 3C) or severely underdeveloped (*asterisk*, Figure 3D) eyes as well as acutely delayed jaw (*red arrow*) and pharyngeal arch (*orange arrow*) development (Figure 3C,D). Notably, the adverse effects on ocular and craniofacial development were more extensive with the K18a.1 knockdown than with the K8 knockdown. Nevertheless, the observed phenotypes in either case were reminiscent of the craniofacial and ocular defects observed in neurocristopathies caused by the abnormal specification, migration, or differentiation of neural crest cells [43,44,45]. 

Based on these results, we further analyzed the effects of K8 and K18a.1 MO knockdown on neural crest cell dynamics in transgenic GFP reporter zebrafish lines during embryonic craniofacial and ocular development. Lateral live imaging of uninjected and standard control-injected embryos at 48 hpf revealed spatially distinct neural crest cell subpopulations, where *sox10*-GFP-positive cells were primarily detected in the craniofacial region (*solid arrows*, Figure 4A,B), with very few cells in the anterior segment (solid and dashed circles), while *foxd3* GFP-positive cells were primarily observed migrating through the optic fissure (*white open arrow*, Figure 4A′,B′) into the anterior segment. A considerable disruption of cranial neural crest cell migration was observed in *krt8* and *krt18a.1* MO-injected Tg(-*4.7sox10:*:EGFP) (Figure 4C,D) and Tg(*foxd3*::GFP) (Figure 4C′,D′) fish. Additionally, both *sox10*- and *foxd3*-positive subpopulations were severely dysregulated and confined to the periocular region with little to no migration into the developing eye.

While Foxd3 and Sox10 are vital embryonic transcription factors for the specification and development of the neural crest, retinal homeobox protein 3 (Rx3), expressed in the early primordia during eye development, is essential for the determination of retinal cell fate and the regulation of stem cell proliferation. MO-mediated K8 and K18a.1 knockdown resulted in a significant loss of *rx3*, *sox10*, and *foxd3* promoter-driven GFP expression (Figure 5C–C″,D–D″,E) in 48-hpf Tg(*rx3*::GFP), Tg(-*4.7sox10*::EGFP), and Tg(*foxd3*::GFP) zebrafish embryos compared with that in uninjected and control MO-injected fish (Figure 5A–A″,B–B″). We also observed substantial neural crest cell disorganization in the optic cup rim, which is the precursor to the neural crest cell-derived iris and ciliary body (*yellow box*, Figure 5C,D), and in the periocular region (*magenta arrow*, Figure 5C′,D′,C″,D″), and anterior segment (*orange box*, Figure 5C′,D′,C″,D″), in MO-injected fish. Interestingly, in all cases, the observed defects were again more profound with the K18a.1 knockdown than with the K8 knockdown. Thus, K18a.1 and, to a lesser extent, K8 played an important role in the migration and patterning of neural crest cells in the eye and craniofacial region during early development.

### 3.4. Retinoic Acid Regulates Keratin Intermediate Filament Gene Expression in the Ocular and Craniofacial Neural Crest during Embryonic Development

The signaling molecule RA is a major regulator of cranial neural crest cell subpopulations during embryogenesis [8,46,47,48,49,50]. As such, the influence of RA is dependent on coordinated, finely tuned, and highly dynamic signaling pathways that mediate the action of this morphogen in both temporal and spatial dimensions. Therefore, identifying the multiple downstream targets of RA signaling in the neural crest is paramount to understanding the contributions of these cells to anterior segment development. To examine whether changes in RA levels influence *krt8/18a.1* expression in embryonic neural crest cells, we performed chromogenic (brightfield, *blue*) in situ hybridization analyses of 48-hpf Casper zebrafish embryos (~1200 embryos total) following exogenous treatment at 24 hpf with 10 μM N,N-diethylaminobenzaldehyde (DEAB), a selective pan-aldehyde dehydrogenase inhibitor of endogenous RA synthesis [51], or at 27 hpf with 100 nM ATRA, an active vitamin A metabolite and RA receptor ligand [17,24,47]. Phenotypic analysis confirmed that compared to untreated animals, treatment with DMSO (Figure 6A,B,E,F,I) did not significantly affect ocular and craniofacial structures. In contrast, and as expected [14,24,47,52], zebrafish subjected to both exogenous treatments showed effects consistent with congenital craniofacial and ocular anomalies, including significantly decreased eye size (*dashed and solid circles*, Figure 6C,D,G,H,I), as measured along the dorsoventral and anteroposterior axes, and markedly retarded jaw and pharyngeal arch formation (*red and orange arrows*, respectively, Figure 6C,D,G,H). 

Interestingly, dysregulation of RA levels in response to exogenous pharmacological treatments decreased the expression of *krt18a.1* but not that of *krt8* in the anterior chamber (*orange dashed box*) and hyaloid (*yellow dashed box*) regions of the eyes of treated zebrafish at 48 hpf (Figure 7B,C,E–G) compared with those of their age-matched DMSO control-treated counterparts (Figure 7A,D,G). These results suggest that *krt18a.1*, but not *krt8*, is a downstream target of RA regulation in the cranial neural crest during early embryonic development.

### 3.5. K8/18a.1 Is Expressed at Various Time Points during Postnatal Corneal Wound Healing

We next examined whether the observed embryonic ocular K8 and K18a.1 expression persists postnatally and during recovery from ocular insult, a circumstance likely requiring the recapitulation of developmental processes to repair the damaged tissues. To this end, the anterior segment of the zebrafish eye was used as a model to evaluate the expression of K8 and K18a.1 in neural crest-derived adult ocular tissues and assess their involvement during postinjury ocular regeneration. For these experiments, we employed a full-thickness corneal injury model involving epithelial debridement, Bowman’s membrane removal, and anterior-to-mid stroma excavation using a corneal burr (Algerbrush II; ~24 adult fish total). Fluorescein dye tracing to distinguish injured corneal surfaces on whole zebrafish eyes (Figure 8A–F) showed that the injured eye was completely healed by 24 h postinjury (hpi) (Figure 8F). Immunohistochemical analysis of 10 μm sections of the uninjured eye revealed corneal epithelial, stromal, and endothelial K8/18a.1 expression (Figure 8G), which was immediately disrupted upon ocular injury (0 hpi, Figure 8H,M). The corneal K8 and K18a.1 expression was restored with the re-establishment of the corneal epithelium at 1 hpi (Figure 8I,M) and remained until the complete regeneration of the cornea at 24 hpi (Figure 8L,M), suggesting the involvement of both K8 and K18a.1 in the maintenance and regeneration of the adult cornea in zebrafish. Notably, quantification analysis showed differential K8 and K18a.1 expression patterns during corneal regeneration from 1 to 24 hpi. The expression of K8 peaked at 1 hpi and then markedly decreased with increasing time postinjury to levels significantly lower than those detected prior to ocular injury (Figure 8M). In contrast, K18a1 expression initially peaked at 1 hpi, sharply decreased at 4 hpi, and then gradually increased thereafter to levels consistent with those detected prior to ocular injury (Figure 8M). Additionally, K8 expression was primarily observed in the corneal endothelium and stroma, whereas K18a.1 expression was observed in all three corneal layers (epithelium, stroma, and endothelium). Thus, K8 and K18a.1 played important but perhaps different spatially and temporally regulated roles in the maintenance and repair of postnatal neural crest-derived ocular tissues, such as the cornea, in adult zebrafish.

### 3.6. Retinoic Acid Regulates Corneal Regeneration and Alters K18a.1 Expression in the Adult Zebrafish Eye during Wound Healing

The effects of alterations in RA levels on postnatal corneal wound healing and K8/18a.1 expression at 24 hpi in adult zebrafish treated with or without 0.1% DMSO, 10 mM DEAB, or 100 nM ATRA (~104 adult fish total) were analyzed. Fluorescein dye tracing revealed a disruption in ocular wound healing at 24 hpi in response to alterations in the levels of RA (Figure 9E,F), as the eyes of the DEAB- and ATRA-treated fish at 24 hpi still showed considerable fluorescein staining consistent with corneal abrasion (Figure 9E,F) in contrast to those of the untreated control, which showed the complete exclusion of fluorescein dye at 24 hpi (Figure 9C). Thin section (10 μm) analysis further revealed a loss of endothelial architecture as well as stromal (Figure 9K) and epithelial edema (Figure 9L) in response to the inhibition of RA synthesis via treatment with DEAB and the increased expression of RA via treatment with exogenous ATRA, respectively. Importantly, the eyes of adult zebrafish treated with the 0.1% DMSO vehicle control showed incomplete fluorescein exclusion (Figure 9D) and a slight disorganization of the healed corneal endothelium at 24 hpi (Figure 9J). However, this effect did not seem to interfere with healing, as the other corneal tissues (epithelium and stroma) appeared normal and structurally intact as compared with those in the untreated control. Consistent with the phenotypic observation of the slight influence of DMSO on ocular wound healing, injured adult zebrafish recovered in water containing the vehicle control alone showed decreased K8/K18a.1 expression in the healed cornea at 24 hpi compared to that in the corneas of their untreated counterparts measured at the same postinjury timepoint (dashed lines, Figure 9M). However, compared with the uninjured fish (solid lines, Figure 9M), K8/18a.1 expression levels in the healed corneas of DMSO-treated adult zebrafish were congruent with the results of the previous experiment (Figure 8), where a significant decrease in K8 expression and a restoration of K18a.1 expression to the initial pre-injury levels was observed at 24 hpi. Therefore, to understand the full effect of treatment with DEAB and ATRA on K8 and K18a.1 expression during corneal wound healing, the protein levels detected in response to pharmacological insult were compared to those detected in response to treatment with the vehicle alone (Figure 9N). The results showed that K18a.1 expression was significantly decreased in response to alterations in RA levels, whereas K8 expression was unaffected. Thus, altogether these results further demonstrated the importance of RA signaling during corneal injury repair and validated K18a.1, but not K8, as a downstream target of RA in the neural crest.

## 4. Discussion

Understanding neural crest contributions to ophthalmic development could be key to establishing successful regeneration strategies for their postnatal derivative tissues with poor healing potential. The extraordinary plasticity displayed by neural crest cells during anterior segment formation may be retained into adulthood for the restoration of the structure and function of damaged tissues. However, the specification of the ocular neural crest remains a significant unaddressed problem. Thus, to fully appreciate the ophthalmic regenerative potential of these cells, the current challenge is to determine the molecular events underlying proper ocular neural crest development. To this end, the present study examined the role of keratin intermediate filament proteins K8 and K18a.1 in the developing neural crest during eye morphogenesis and in postnatal corneal tissues during injury repair using embryonic and adult zebrafish models. 

### 4.1. Keratins as Hallmarks of Maturation in Neural Crest-Derived Ocular Tissues

Keratin IFs are a class of cytoskeletal proteins with distinct morphology and distribution in adult tissues (reviewed in [38,53,54,55,56,57]). In particular, the cytokeratins are a family of polypeptides typically found in simple and complex adult epithelial tissues and epithelia-derived cell lines during development [38]. During ocular development, the corneal epithelium derives from surface ectoderm cells adjacent to both sides of the lens placode [58], and studies in humans, mice, rabbits, and chicks have shown that ocular cytokeratin expression typically indicates corneal epithelial cell maturation status [59,60,61]. Indeed, early studies in developing Japanese quail embryos did not detect the distribution of cytokeratins in the neural tube, neural plate, or neural crest during morphogenesis [62]. However, subsequent studies have shown that during embryonic development, cells in hybrid mesenchymal/epithelial differentiation states, such as the migrating neural crest, show keratin intermediate filament protein expression profiles consistent with those of both mesenchymal and epithelial tissues [63,64]. These findings correlate with the keratin subunit (K1, K5, K8, K10, K14, K16, and K18) expression observed in human cultured melanocytes, keratinocytes, and mouse neural crest cells [65,66,67]. Additionally, the heterogeneous expression of cytokeratins 7, 8, 18, and 19 has been demonstrated in multiple studies of human corneal endothelial cells [68,69,70,71]. In the present study, we further demonstrated keratin expression as a hallmark of the development of neural crest-derived ocular tissues. Indeed, *krt8* and *krt18a.1* expression was detected at the neural plate border and in neural crest cells in the periocular and ocular regions during early development in zebrafish. These distinct neural crest subpopulations contribute to the neural crest-derived tissues of the vertebrate eye, including the corneal endothelium and stroma, iris stroma, ciliary body, trabecular meshwork, and sclera (reviewed in [8,9,72,73,74,75]). 

### 4.2. Keratins as Cranial Neural Crest Cell Fate Signaling Molecules 

As a stem cell population, the neural crest requires the ability to detect a variety of signals at specific times during development for their successful migration, proliferation, and subsequent differentiation into diverse sets of tissues and cells. Deficiencies in neural crest cell specification and migration impair optic cup formation and anterior segment development, with phenotypic effects of severe eye abnormalities [7,13]. In zebrafish, the migration of neural crest cells is associated with the differentiation of their derivative tissues in the ocular anterior segment [76,77]. However, the signals and mechanotransducers required for terminal neural crest cell differentiation towards tissues and cell fates in the anterior segment are less well understood. A recent study showed that keratin intermediate filaments function as fate determinants in mammalian embryonic development [78]. In the present study, we showed that *krt8* and *krt18a.1* play a role in neural crest specification and migration in the developing anterior segment and craniofacial region. Knockdown of both K8 and K18a.1 significantly affected the migratory dynamics of neural crest cells in the developing zebrafish eye, and aberrant ocular defects, including coloboma, microphthalmia, anophthalmia, and jaw malformation, were observed. While these defects reflect the abnormal specification, migration, or differentiation of neural crest cells, it is worth noting that, unlike their actin filament and microtubule cytoskeletal counterparts, intermediate filaments do not facilitate cell movements but rather appear to exert mechanical forces and intercellular interactions that may regulate cell fate decisions [79]. Thus, additional studies are needed to determine whether this influence could be achieved through direct interactions with signaling proteins or by regulating the localization of specific signaling molecules. Nevertheless, these studies demonstrated the essentiality of *krt8* and *krt18a.1* for neural crest development and anterior segment formation.

### 4.3. Keratins as Downstream Targets of Retinoic Acid in the Ocular Anterior Segment

The vitamin A metabolite RA is a potent regulator required for the development of all higher vertebrates from fish to humans. With respect to ocular development, cranial neural crest cells originating from the prosencephalon, diencephalon, and anterior mesencephalon require the tight control of RA levels at multiple steps during development for their precise survival, migration, and eventual differentiation into structures in the midline of the face and anterior segment of the eye [8,9,24,46,47,48,49,50,80,81,82,83,84]. In the present study, we identified *krt18a.1* as a downstream target of RA regulation during anterior segment formation in the embryonic zebrafish eye. This result is consistent with the impact of RA signaling on the developing ocular neural crest and the involvement of keratin intermediate filament genes in that process. Interestingly, both increased and decreased RA downregulated *krt18a.1* expression. Previous studies have shown that increased RA expression (ATRA) decreases cell survival and inhibits ventral cranial neural crest cell migration, while decreased RA expression (DEAB) markedly disrupts both dorsal and ventral cranial neural crest cell migration, resulting in congenital malformations, such as cleft lip and/or palate and ocular anterior segment dysgenesis [24,47]. Herein, we observed dorsal and ventral patterning of both *krt8* and *krt18a.1* in the developing zebrafish. Accordingly, following pharmacological insult, the universal decreased expression of *krt18a.1* in the periocular region and anterior segment of the eye was reasonably expected. Howbeit, the lack of a significant effect on its canonical binding partner *krt8* was not expected.

Like that for ocular anterior segment tissues, RA is also a crucial regulator of the development and differentiation of epithelial tissues. However, although K8 and K18 are an essential epithelial cell-specific intermediate filament binding pair, little is known about the influence of RA on K8, while that of K18 has been repeatedly demonstrated [85,86]. Notably, the regulation of individual keratin genes is mediated through the binding of transcription factors to nearby DNA elements [86,87,88,89,90,91,92,93]. These transcription factors respond to signaling molecules, such as growth factors, mitogens, hormones, and vitamins, in the extracellular milieu that affect the overall expression of each keratin gene. So, although all known keratins tend to be expressed and act in specific pairs comprising a basic (Type II) and an acidic (Type I) keratin, the corresponding pairwise regulation of their gene expression has not yet been reported. Thus, the differential effect of RA regulation between *krt8* and *krt18a.1* in the cranial neural crest likely reflects independent regulatory mechanisms that ultimately dictate the expression of each keratin gene independently from the other. 

### 4.4. Keratins as Contributors to Corneal Wound Healing and Regeneration 

#### 4.4.1. Zebrafish Models of Corneal Regeneration and Injury Repair

Herein, a zebrafish model of corneal injury was employed to examine the idea that postnatal wound healing recapitulates embryogenesis through the activity of multipotent neural crest stem cells, and as such, the involvement of K8 and K18a.1 should be apparent during injury repair. Zebrafish are excellent models to study regeneration and injury repair processes, as their eyes are very similar to those of their human counterparts. Notwithstanding, in contrast to humans, zebrafish can avoid scarring and fibrosis and successfully regenerate damaged tissues [5,25,94,95,96,97,98,99,100]. In this study, we observed the complete regeneration of the adult zebrafish eye by 24 hpi, with the re-establishment of the corneal layers within 1 h after ocular insult. 

#### 4.4.2. Keratin Contributions to Corneal Wound Healing

Similar to that detected during anterior segment formation, K8 and K18a.1 expression was also detected during the maintenance and regeneration of neural crest-derived corneal tissues in the adult zebrafish eye. Interestingly, although equally expressed prior to injury, these intermediate filament proteins showed differential and inverse expression patterns during injury repair, with increased K8 and decreased K18a.1 expression in the first 4 h postinjury, and decreased K8 and increased K18a.1 expression thereafter. Healing from full-thickness corneal injury, involving extensive damage to the corneal epithelium and underlying stroma and/or endothelium, is a complex process requiring the immediate apoptosis and subsequent activation, proliferation, migration, and transdifferentiation of neural crest-derived corneal cells, i.e., keratocytes and endothelial cells, to eventually restore the structure, transparency, and function of the injured cornea [101,102,103]. As such, the differential keratin expression observed in zebrafish corneal wound healing likely reflects the induction and suppression of these genes with the progression of healing. In humans, upregulated K8/18 expression promotes tumor progression and is predictive of a poor prognosis in multiple cancers: increased KRT8 expression induces cell apoptosis and promotes cell proliferation, cell migration, invasion, and EMT [104,105,106], while increased KRT18 expression is associated with the increased viability, migration, and invasion of cancer cells and the selective regulation of cell proliferation and apoptotic genes [107,108,109]. 

#### 4.4.3. Effects of Retinoic Acid on Keratin Expression during Wound Healing

Consistent with the effects observed during ocular embryogenesis, alterations in RA levels only affected K18a.1 expression, which was significantly downregulated during corneal wound healing in response to treatment with both DEAB and ATRA. Consequently, severe corneal epithelial/stromal edema and disrupted corneal endothelial reformation were also observed. Although these results are consistent with the role of RA as a morphogenic regulator of key signaling molecules that affect keratin expression during both embryonic development and tissue regeneration [5,17,46,49,81,110,111,112,113,114,115,116,117,118,119,120,121,122], they are in direct conflict with the results of previous studies showing a positive effect of RA on corneal wound healing by suppressing deleterious events, such as corneal opacity and neovascularization [123,124,125]. This discrepancy may reflect a dosage effect of RA in the cornea during regeneration independent of its regulation of *krt18a.1* and beyond the scope of the present study. Nevertheless, we have demonstrated the participation of *krt8* and *krt18a.1* in the injury repair of postnatal neural crest-derived corneal tissues.

## 5. Conclusions

In the present study, we examined the roles of *krt8* and *krt18a.1*, as additional factors expressed in the neural crest during early ocular development in zebrafish. The contributions of both genes to ocular anterior segment development and corneal wound healing altogether suggest a permissive role for these intermediate filament proteins in the coordinated migration, proliferation, and survival of neural crest cells during the embryogenesis, postnatal maintenance, and postinjury restoration of neural crest-derived tissues in the eye.

## Figures and Tables

**Figure 1 cells-13-01473-f001:**
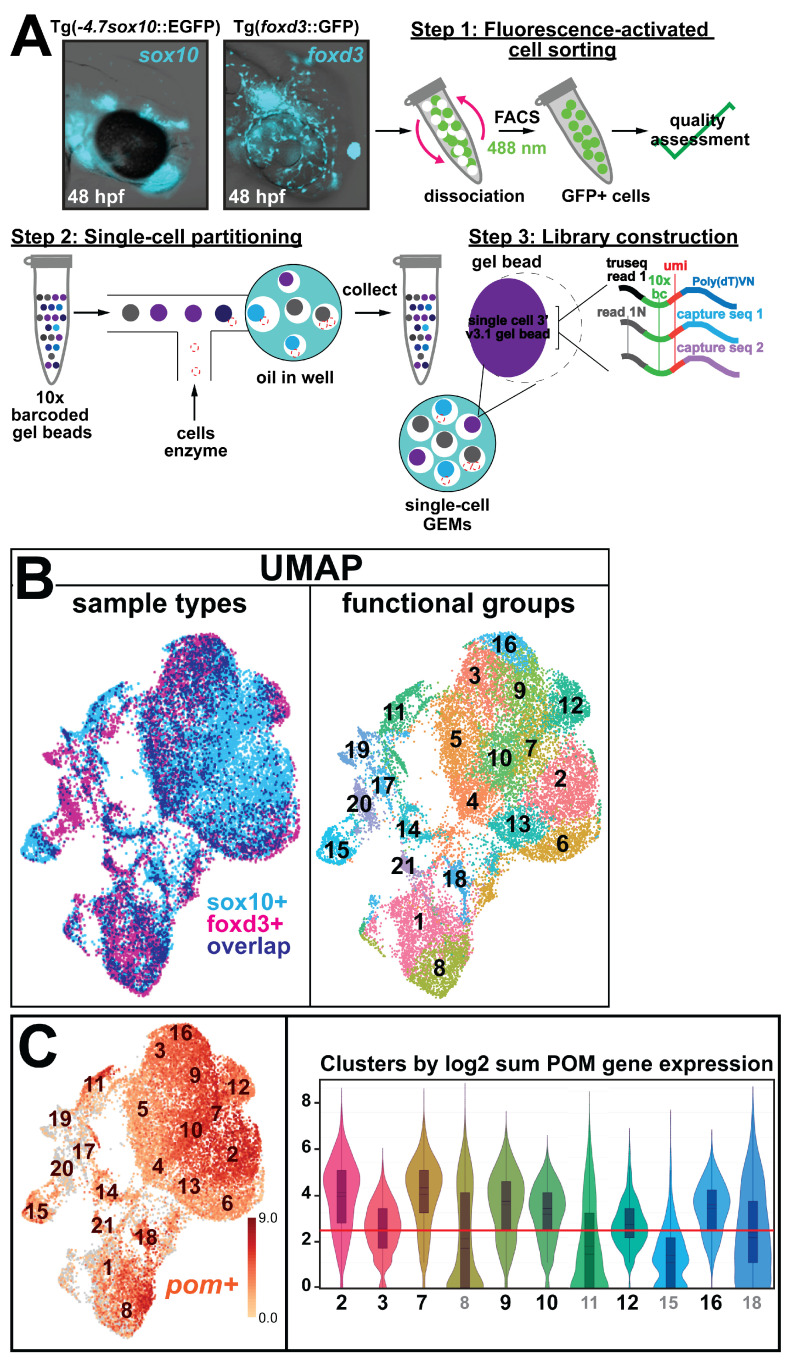
Keratin intermediate filament genes are expressed during early ocular development. (**A**) Step 1: Live images (lateral views) of the heads of 48-hpf Tg(-*4.7sox10*::EGFP) and Tg(*foxd3*::GFP) embryos for the collection of distinct *sox10* (left panel) and *foxd3* (right panel) POM neural crest cell subpopulations by FACS to isolate the GFP+ cells. Step 2: Following quality checks, the samples were loaded onto the 10× Genomics Chromium platform for the partitioning and encapsulation of single cells into nanoliter-sized GEMs (Gel beads-in-EMulsion). Each encapsulated cell was then lysed within its GEM, and the released RNA was reverse transcribed to cDNA with primers attached to a gel bead carrying a unique 10× barcode for downstream cell separation. Subsequently, the GEMs were broken, and all uniquely barcoded cDNAs were pooled, followed by PCR to generate enough material. Step 3: Illumina sequencing and library construction were performed using the 10× Genomics Single Cell 3′ v3.1 Protocol. Raw sequencing data were demultiplexed and the FASTQ files were aligned to the zebrafish reference genome (danRer11) using Cell Ranger. Loupe Browser (version 7.0.1) was used to evaluate the data and for further downstream analysis. (**B**) Cluster distribution at 48 hpf. *Sox10*:GFP and *foxd3*:GFP periocular neural crest cell subpopulations primarily clustered together, with few non-overlapping *sox10*+ and *foxd3*+ clusters (left panel). In total, 21 functional subgroups were observed in the combined dataset (right panel). (**C**) Cell type analysis based on the high expression of periocular mesenchyme (POM) transcription factors previously implicated in regulating ocular neural crest cell migration and differentiation (*eya2*, *foxc1a*, *foxc1b*, *foxo1a*, *lmx1ba*, *lmx1bb*, *pitx2*, *tfap2a*, and *tfap2b*), revealed 7 functional subgroups of interest. (**D**) Pathway enrichment analysis (Metascape) identified common and distinct putative biological functions (*black* and *gray font*) represented by the genes in the 7 functional subgroups of interest. In addition, healing and repair processes (*blue font*) were also highlighted amongst these functional subgroups. Moreover, the functional subgroup (Cluster 9) associated with ‘eye morphogenesis’, ‘ECM remodeling’, and ‘regeneration’ was also related to ‘intermediate filament (IF)-based processes’ (*magenta font*). (**E**) STRING analysis identified intermediate filament gene networks comprising *krt94*, *krt8*, *krt18a.1*, and *krt18b* among clusters associated with ‘eye morphogenesis’ and ‘neural crest development’ (Clusters 9, 12, 16). (**F**) Intermediate filament binding partners *krt8* and *krt18a.1/18b* showed marked upregulated expression (at least 2.5x, as indicated by the *red lines*) during anterior segment development.

**Figure 2 cells-13-01473-f002:**
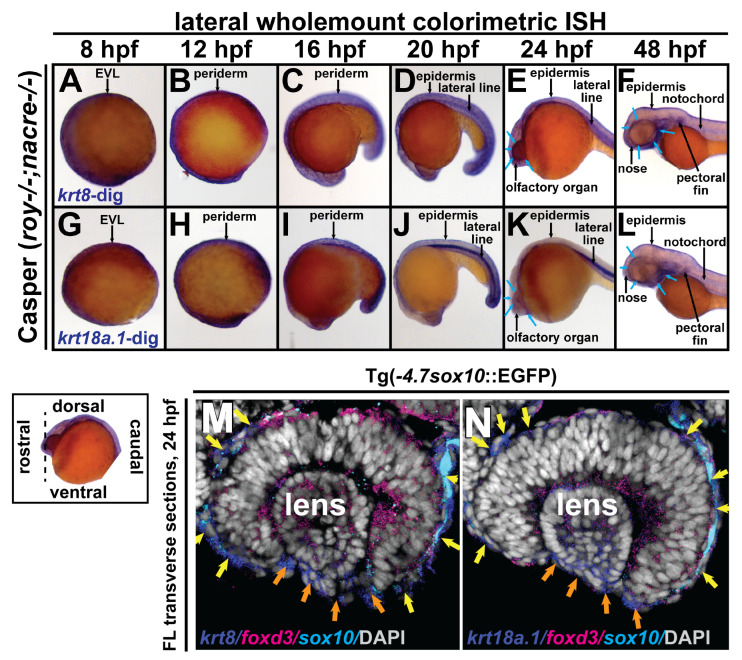
*Krt8*/*krt18a.1* are expressed in the ocular and periocular neural crest during early development. Wholemount in situ hybridization in Casper (*roy*-/-; *nacre*-/-) zebrafish embryos during early development at 8, 12, 16, 20, 24, and 48 hpf. K8 and K18a.1 gene expression was detected using a colorimetric assay (Vector Blue Substrate Kit, Vector Laboratories) that is both chromogenic (*blue*) and fluorescent (Cy5). The sections were mounted in a media containing DAPI (*gray*). Lateral brightfield wholemount images show *krt8* (**A**–**F**) and *krt18a.1* (**G**–**L**) expression initiating dorsally along the neural plate border and enveloping layer (EVL) at 8 hpf (**A**,**G**) then dorsoposteriorally in the embryonic epithelium and ventrally in the ocular and craniofacial regions (**B**–**E**,**H**–**K**), with apparent expression in the ocular anterior segment and facial mesenchyme (*blue arrows*, (**F**,**L**)) by 48 hpf. Fluorescent double in situ hybridization for *krt8* (**M**) or *krt18a.1* (**N**) (Cy5/*dark blue*) and *foxd3* (Texas Red/*magenta*) expression, followed by immunohistochemistry for *sox10*-GFP (α-GFP/*light blue*) was performed on transverse cephalic sections. The black dashed line (*lower left* insert) indicates the orientation of the plane of section, which passes perpendicular to the spinal column and extends in the rostral-caudal direction. Fluorescent confocal microscopy revealed *krt8/krt18a.1* expression in the neural crest-derived ocular anterior segment (primordial cornea, *orange arrows*) and periocular mesenchyme (*yellow arrows*) of zebrafish embryos at 24 hpf.

**Figure 3 cells-13-01473-f003:**
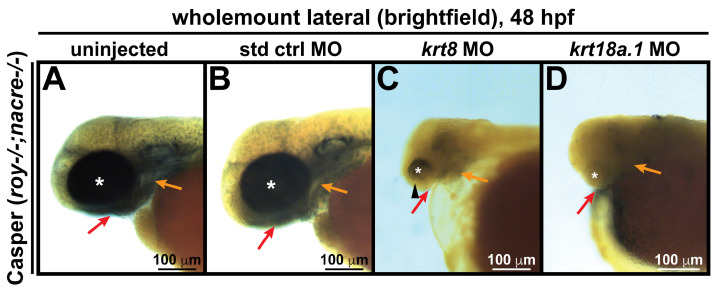
Krt8/Krt18a.1 knockdown disrupts ocular and craniofacial development. Wholemount imaging showed that compared with their uninjected (**A**) and standard control (std ctrl) MO-injected (**B**) counterparts, *krt8* MO-injected zebrafish embryos (**C**) developed microphthalmic eyes (*asterisk*) with coloboma (*arrowhead*), while *krt18a.1* MO-injected embryos (**D**) had severely underdeveloped or anophthalmic eyes (*asterisk*) at 48 hpf, suggesting that the effects were more severe with Krt18a.1 knockdown. The absence of pharyngeal arch (*orange arrows*) and jaw (*red arrows*) formation was observed after MO knockdown in both cases.

**Figure 4 cells-13-01473-f004:**
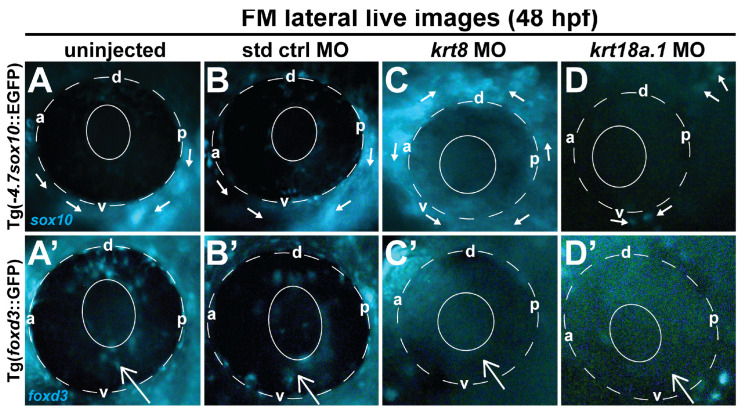
Krt8/Krt18a.1 MO knockdown disrupted ocular neural crest cell migration dynamics during early eye development. Lateral live imaging at 48 hpf shows the effects of Krt8 and Krt18a.1 knockdown on GFP reporter expression in Tg(*foxd3*::EGFP) and Tg(-*4.7sox10*::EGFP) zebrafish embryos injected at the single-cell stage with antisense MOs targeting *krt8* and *krt18a.1*. (**A**,**A′**) uninjected, (**B**,**B′**) standard control (std ctrl) MO-injected, (**C**,**C′**) krt8 MO-injected, and (**D**,**D′**) krt18a.1 MO-injected. The migration of GFP-positive neural crest cells into the ocular (*open arrow*) and periocular (*solid arrows*) regions was markedly disrupted in response to Krt8/Krt18a.1 MO knockdown compared with that in uninjected and control MO-injected embryos. The solid and dashed circles highlight the lens and retinal pigment epithelium, respectively, of the zebrafish eye. d, dorsal; v, ventral; p, posterior; a, anterior.

**Figure 5 cells-13-01473-f005:**
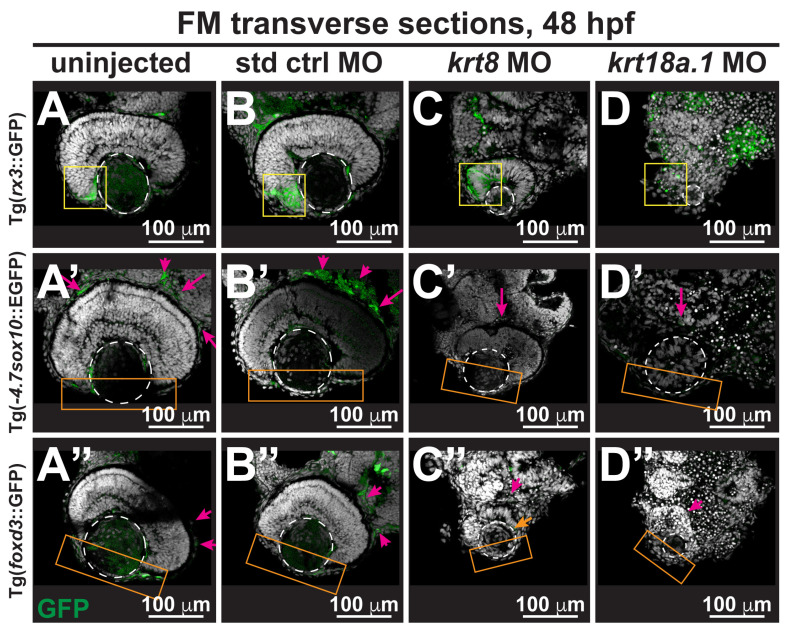
Krt8/Krt18a.1 play roles in neural crest migration and patterning during early eye development. Section analysis of 48-hpf Tg(-*4.7sox10*::EGFP), Tg(f*oxd3*::GFP), and Tg(*rx3*::GFP) zebrafish embryos showed that MO knockdown of K8 or K18a.1 resulted in the significant disorganization of neural crest cells in the optic cup rim (*yellow box*, (**C**,**D**)), periocular mesenchyme (*magenta arrow*, (**C′**,**D′**,**C″**,**D″**)), and ocular anterior segment (*orange box*, (**C′**,**D′**,**C″**,**D″**)). The *orange arrow* indicates slight foxd3 signal in the anterior segment of the underdeveloped eye. The *dashed circles* highlight the lens. (**E**) Quantitative analysis showed a significant decrease of *rx3*, *sox10*, and *foxd3* promoter-driven GFP expression in the periocular and ocular regions of the developing eyes of *krt8/krt18a.1* morphants compared with those of uninjected (**A**–**A″**) and control MO-injected fish (**B**–**B″**). Notably, these effects were more severe with K18a.1 knockdown. **, *p*-value < 0.01; n.s., not significant.

**Figure 6 cells-13-01473-f006:**
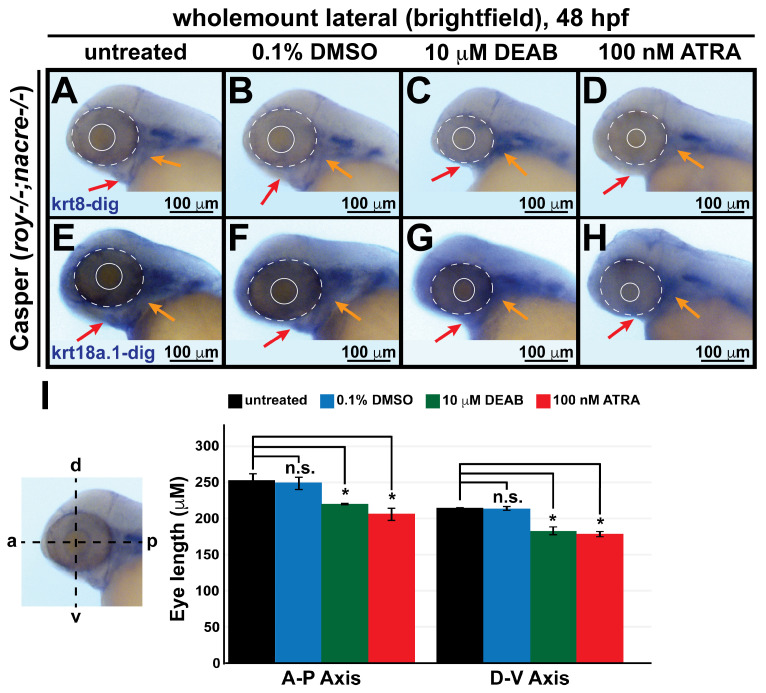
Changes in retinoic acid levels affect early ocular and craniofacial development. Wholemount in situ hybridization was performed using a chromogenic (*blue*) colorimetric assay. Lateral images (**A**–**D**,**E**–**H**) of treated zebrafish embryos taken at 48 hpf show the teratogenic effects of pharmacological insult with 10 μM DEAB and 100 nM ATRA on the ocular and craniofacial neural crest during early development. The solid and dashed circles highlight the reduced eye size. The *red* and *orange arrows* highlight jaw and pharyngeal arch malformation, respectively. (**I**) Quantitative analysis of this effect shows that treatment with both DEAB and ATRA delayed ocular development and significantly decreased the eye size of the treated fish, as measured along the anterior–posterior (a-p) and dorsal-ventral (d-v) axes. *, *p*-value < 0.05; n.s., not significant.

**Figure 7 cells-13-01473-f007:**
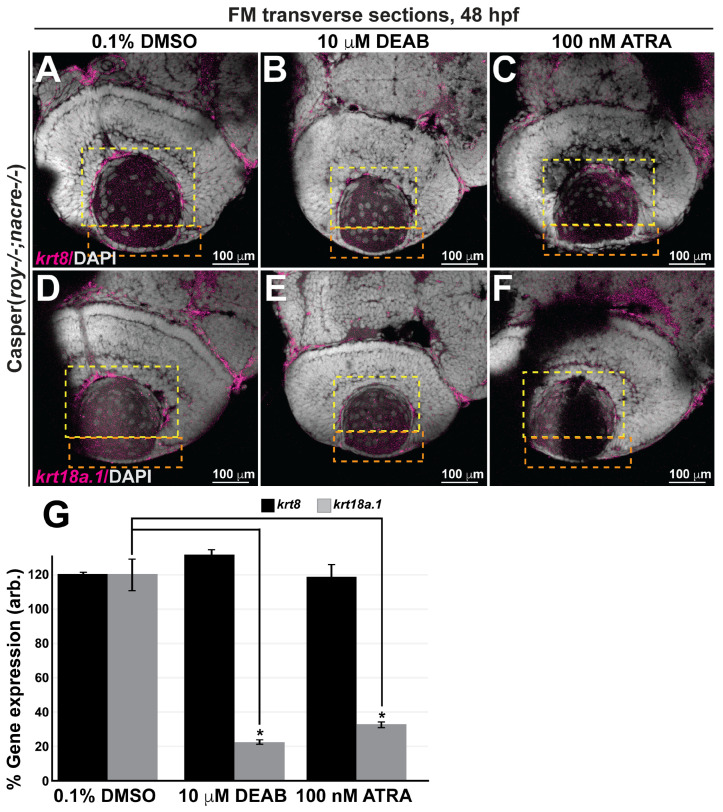
Retinoic acid regulates krt18a.1, but not krt8, expression in the eye during early embryogenesis. (**A**–**F**) Treatment of Casper zebrafish embryos at 24 and 27 hpf with DEAB and RA, respectively, followed by wholemount colorimetric in situ analysis shows decreased *krt18a.1,* but not *krt8*, expression in the developing eye in response to pharmacological insult. (**G**) Quantification of these effects, as measured in the anterior chamber (*orange dashed box*) and hyaloid (*yellow dashed box*) regions of the eyes of 48-hpf zebrafish, further shows that compared with the DMSO vehicle control, treatment with either DEAB or ATRA significantly decreased *krt18a.1* expression, while that of *krt8* was not affected. *, *p*-value < 0.05.

**Figure 8 cells-13-01473-f008:**
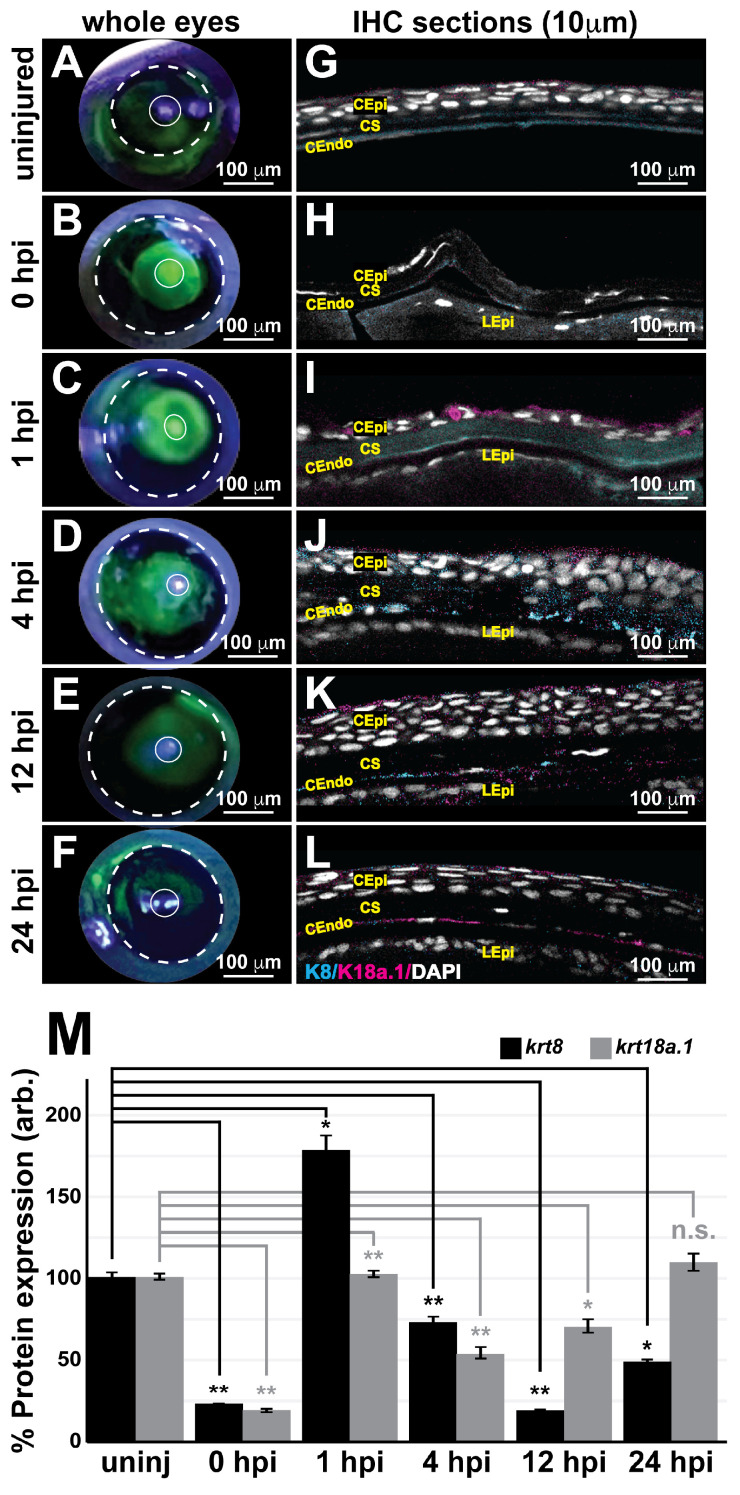
Keratin intermediate filament proteins 8 and 18a.1 are expressed at various time points during postnatal corneal wound healing. A zebrafish adult ocular injury model was established to achieve extensive corneal injury consisting of epithelial debridement, removal of Bowman’s membrane, and excavation of the anterior to mid stroma. (**A**–**F**) Fluorescein tracing to distinguish injured corneal surfaces shows the complete healing of the injured eye by 24 h post injury (hpi). The dashed and solid circles highlight the iris and pupil, respectively, in the zebrafish eye. (**G**–**L**) Immunohistochemical analysis of thin (10 μm) sections shows that the corneal stroma regenerated within 1 h following mechanical injury, and the expression of both K8 and K8a.1 in neural crest-derived tissues (stroma and endothelium) was detected in the uninjured (uninj) cornea and during ocular wound healing. CEpi, corneal epithelium; CS, corneal stroma; CEndo, corneal endothelium. (**M**) Quantitative analysis of K8 and K18a.1 expression shows differential expression patterns during corneal wound healing. *, *p*-value < 0.05; **, *p*-value < 0.01; n.s., not significant.

**Figure 9 cells-13-01473-f009:**
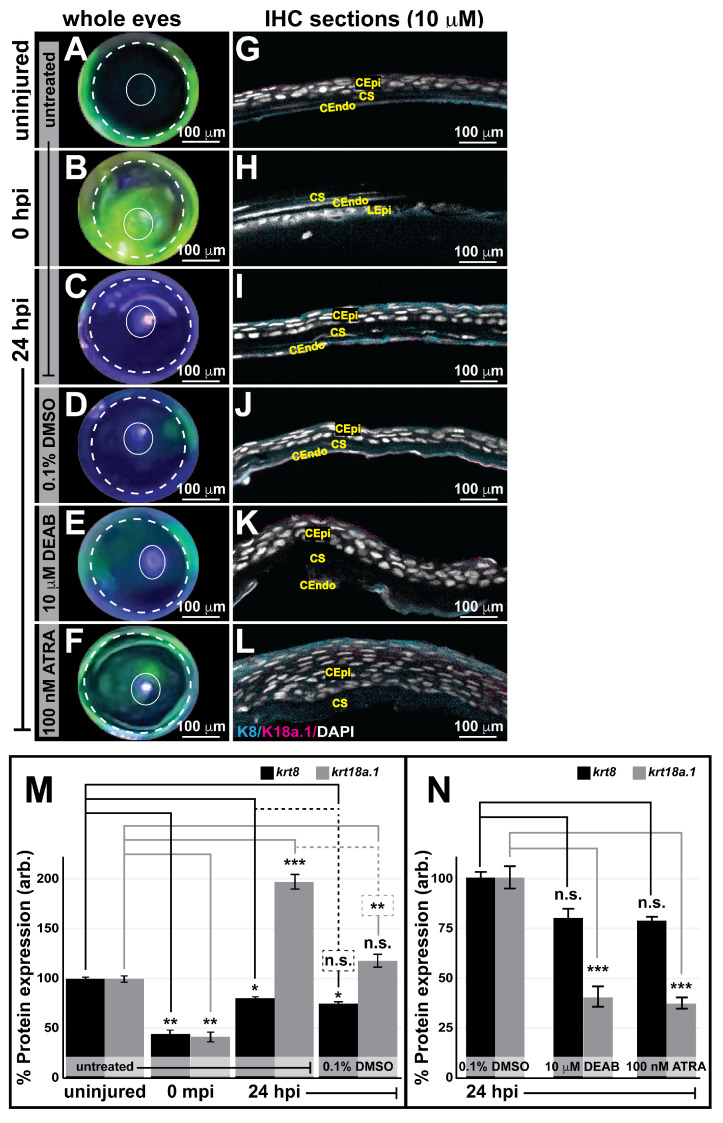
RA influences injury repair and regulates K18a.1 expression in the postnatal neural crest. Adult zebrafish subjected to ocular injury were subsequently recovered in system water containing lidocaine (2.5 mg/L) or lidocaine in combination with either 0.1% DMSO, 10 mM DEAB, or 100 nM ATRA. (**A**–**F**) Fluorescein tracing shows that corneal healing was remarkably disrupted by treatment with DEAB (**E**) and ATRA (**F**). The dashed and solid circles highlight the iris and pupil, respectively, in the zebrafish eye. (**G**–**L**) Cryosection (10 μm), followed by immunohistochemical analysis revealed significant stromal edema (**K**) and epithelial (**L**) edema in response to alterations in RA levels during ocular wound healing. CEpi, corneal epithelium; CS, corneal stroma; CEndo, corneal endothelium. (**M**,**N**) Quantitative analysis of K8 and K18a.1 expression revealed decreased Krt18a.1 expression in response to pharmacological insult during corneal wound healing. *, *p*-value < 0.05; **, *p*-value < 0.01; ***, *p*-value < 0.001; n.s., not significant.

## Data Availability

The data presented in this study are available on request from the corresponding authors. The data are not publicly available, as Ann & Robert H. Lurie Children’s Hospital of Chicago does not require data sharing.

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
