# Peer review of "Keratin 8/18a.1 Expression Influences Embryonic Neural Crest Cell Dynamics and Contributes to Postnatal Corneal Regeneration in Zebrafish"

_cells, 2024, doi:10.3390/cells13171473_

Round 1
Reviewer 1 Report
Comments and Suggestions for Authors
In article “Keratin 8/18a.1 expression regulates embryonic neural crest cell dynamics and plays a role in postnatal ocular wound healing in zebrafish,” the authors presented a comprehensive analysis of studies of the mechanodynamics of neural crest cells during development to assess the value of NSCs in the postembryonic period in the treatment of injury. In general, the work is interesting and includes a variety of approaches for assessing regenerative processes, however, a number of significant comments do not allow publishing this work in its presented form for the following reasons:
1. The Results section does not indicate how many total fish were used in this study.
2. The method for conducting transcriptomic analysis is described too schematically; the main sequencing protocol needs to be specified in more detail.
3. It is necessary to provide the current protocol for in situ hybridization; the authors indicate that this procedure was carried out according to the previously described protocol is not sufficient.
4. The calorimetric evaluation algorithm is not clear. The authors indicate "the development of sensory controls to provide specific staining" but do not demonstrate evidence of such development.
5. The work used cross-adsorbed goat anti-rabbit IgG secondary antibodies Alexa Fluor™ 488 (1:200; Life Technologies, Carlsbad, California, USA). The results of immunolabeling are inconclusive, in Fig. 2 multiple immunohistochemical staining does not demonstrate specific staining. The authors did not provide controls for antibody specificity. The immunohistochemical labeling protocol is unsatisfactory.
6. In the Pharmacological Treatment section, the authors indicate that for all experiments, dose curves were constructed for each type of pharmacological treatment, and final concentrations were selected based on LD50 and consistency of phenotype, but these data are not presented.
7. Immunohistochemical staining with antibodies against K8 (1:100; GTX34663; GeneTex) and antibodies against K18a.1 (1:500; GTX112978; GeneTex) requires the provision of data on the specificity of these antibodies for zebrafish, in particular Western immunoblotting with protein loading control and /or positive control. There is no evidence of antibody specificity in the work; the authors do not indicate at all in which animal the primary antibodies were obtained.
8. There are also many shortcomings in the Results section; in particular, stylistic control of the text of this section of the work is required. For example, using the expression “ad nauseum” when presenting results is unacceptable.
9. The discussion is too speculative; it is necessary to more substantively discuss the results of our own experiments and interpret the data obtained based on literature data. It is recommended to break the discussion into thematic sections.
Comments on the Quality of English Language
correction required
Author Response
- The Results section does not indicate how many total fish were used in this study.
Response: Thank you for this comment. The numbers of fish (embryos and adults) used in this study are indicated in the Methods section as follows:
- Wholemount in situ hybridization and immunostaining:
Line 157- “at 50 embryos/tube per experimental condition”
- Pharmacological treatments:
Line 214 – “Casper embryos, at 50 to 100 embryos per treatment group”
- Morpholino oligonucleotide injections:
Line 237 – “Each experiment was conducted a minimum of 3 times with at least 25 to 50 embryos per group”
- Corneal abrasion and immunohistochemistry:
Line 247 – “Each experiment was repeated 3-4 times, harvesting 4-6 fish at each collection period”
- The method for conducting transcriptomic analysis is described too schematically; the main sequencing protocol needs to be specified in more detail.
Response: Thank you for this comment. We have consulted with the Northwestern University Sequencing Core (NUSeq), and the protocol for conducting the transcriptomics analysis has been rewritten in more detail as requested.
- It is necessary to provide the current protocol for in situ hybridization; the authors indicate that this procedure was carried out according to the previously described protocol is not sufficient.
Response: Thank you for this comment. The current protocol for in situ hybridization has now been described as requested in the revised version of this manuscript.
- The calorimetric evaluation algorithm is not clear. The authors indicate "the development of sensory controls to provide specific staining" but do not demonstrate evidence of such development.
Response: Thank you for this comment. To clarify, the in situ hybridization was developed using a colorimetric analysis, that is, an assay in which a color (i.e., blue) is formed during the reaction of a detection chemical (i.e., vector blue or vector red; Vector Labs) with the target substance (i.e., alkaline phosphatase). Moreover, “sensory controls” were not developed, but rather “sense RNA probe control experiments were also performed and developed (data not shown) in parallel to ensure specific antisense RNA probe binding”. Because the sense RNA probe control experiments resulted in no staining (a negative result), these data are not shown here. To prevent further confusion and provide clarification, this description has been rewritten in the text of the revised document.
- The work used cross-adsorbed goat anti-rabbit IgG secondary antibodies Alexa Fluor™ 488 (1:200; Life Technologies, Carlsbad, California, USA). The results of immunolabeling are inconclusive, in Fig. 2 multiple immunohistochemical staining does not demonstrate specific staining. The authors did not provide controls for antibody specificity. The immunohistochemical labeling protocol is unsatisfactory.
Response: Thank you for this comment. To clarify, Figure 2 shows the results of a double in situ hybridization experiment in Casper Tg(-4.7sox10:GFP) embryos using specific antisense RNA probes targeting krt8 or krt18a.1 (digoxigenin-labeled) and foxd3 (fluorescein-labeled), identified using alkaline phosphatase-conjugated anti-dig and anti-flour antibodies, respectively, and subsequently developed with vector blue and vector red detection kits, respectively. These detection kits are both chromogenic (blue or red, respectively) and fluorescent (Cy5 or Texas Red, respectively). To amplify the GFP signal of the transgenic fish, in situ hybridization was followed by wholemount immunohistochemistry using an anti-GFP antibody and Alexa Fluor™ 488 secondary antibody. The specificity of the GFP antibody used (ab6556; Abcam) has previously been demonstrated by the manufacturer and in many of our previous papers (Chawla, et al., 2016; Eason, et al., 2071; Williams, et al., 2017 and 2022). Images capturing both the in situ hybridization and immunostaining results were obtained using fluorescence microscopy as described in the results section. We apologize for the confusion and have rewritten this description for clarification in the Methods, Results and figure legend (Figure 2) of the revised document. The immunohistochemical staining protocol has also been rewritten to provide the procedural details as requested.
- In the Pharmacological Treatment section, the authors indicate that for all experiments, dose curves were constructed for each type of pharmacological treatment, and final concentrations were selected based on LD50 and consistency of phenotype, but these data are not presented.
Response: Thank you for this comment. The data corresponding to the dose curves performed for RA and DEAB is not shown, as we do not consider these data directly relevant to the main topic of the paper. Moreover, the final concentrations used (i.e., 100 nM RA and 10 μM DEAB) are consistent with the concentrations used in previous published studies resulting in the same phenotype (Williams et al., 2022). This information has been added to the text in this section of the revised manuscript. Notably, if requested, the dose curve information can be provided as supplementary material.
- Immunohistochemical staining with antibodies against K8 (1:100; GTX34663; GeneTex) and antibodies against K18a.1 (1:500; GTX112978; GeneTex) requires the provision of data on the specificity of these antibodies for zebrafish, in particular Western immunoblotting with protein loading control and /or positive control. There is no evidence of antibody specificity in the work; the authors do not indicate at all in which animal the primary antibodies were obtained.
Response: Thank you for this comment. We agree that the specificity of commercial (GeneTex) monoclonal (mouse) K8 (1:100; GTX34663; GeneTex) and polyclonal (rabbit) K18a.1 (1:500; GTX112978; GeneTex) antibodies, both reported to show reactivity to zebrafish according to the manufacturer’s specifications, were not further validated by us. Following the institution’s recent cybersecurity event, we have just regained the ability to purchase recombinant proteins for both keratins. Notably, these items are on backorder. However, once we receive these items, we can perform WB analyses of zebrafish lysates to demonstrate antibody specificity. If requested, the results of the WB analyses can be provided as supplementary material.
- There are also many shortcomings in the Results section; in particular, stylistic control of the text of this section of the work is required. For example, using the expression “ad nauseum” when presenting results is unacceptable.
Response: Thank you for this comment. The Results section has been edited to improve stylistic control of the text. Particularly, the expression “ad nauseum” has been deleted.
- The discussion is too speculative; it is necessary to more substantively discuss the results of our own experiments and interpret the data obtained based on literature data. It is recommended to break the discussion into thematic sections.
Response: Thank you for this comment. The Discussion section has been rewritten to reduce speculation and improve the interpretation of the obtained data based on the current literature. We agree with the reviewer and have broken the discussion into thematic sections.
Reviewer 2 Report
Comments and Suggestions for Authors
This manuscript titled “Keratin 8/18a.1 expression regulates embryonic neural crest cell dynamics and plays a role in postnatal ocular wound healing in zebrafish” investigates the role of keratin intermediate filament genes during early ocular neural crest cell (NCC) development. The authors looked at spatial and temporal expression pattern of these genes, followed them using live imaging, and perturbed these factors to understand their role in NCC migration during homeostasis and disease. Overall, this is a well-written and very well put together manuscript that I would be happy to recommend for publication once these following changes enlisted below are addressed.
Major Comments:
1. Line 205-206: Any particular reason why K-Means was used here instead of KNN associated clustering analysis? K-Means will always provide a user-defined number of clusters from scRNAseq data which may not align with biological data. It may be possible that in this manuscript, as shown in Figure 1B, there may be more or less heterogeneity within either of the sox10:EGFP and foxd3:EGFP cells. Maybe try clustering the data using Seurat that uses a shared nearest neighbor approach to pull transcriptionally similar cells together?
2. Line 207-208: “Most of the genes isolated from the Tg(-4.7sox10::EGFP) (52.3%) and 207 Tg(foxd3::GFP) (44.9%) reporter lines were found in Clusters 1 and 2 (FIGURE 1C)” – how were the number of genes quantified here? Was a threshold of expression used to count genes that are expressed above or below a threshold? What would it mean for the other clusters except Clusters 1 and 2 to have lesser number of genes expressed? Is it because there are lesser cells in those clusters or that the data there is not of high quality? Can the authors please explain why having more genes in just 2 clusters is relevant here?
3. Line 208-211: The authors show that there are several genes that are differentially expressed between sox10:EGFP and foxd3:EGFP populations in addition to the RA signaling and ASD genes. What about similarities between the two populations? As both label neural crest, it would be interesting to see how similar are these two populations? Can the authors please cluster both datasets together and see if sox10:GFP and foxd3:GFP populations cluster together or do they remain as discrete populations?
4. In general, it would also be useful to annotate the 10 clusters per dataset or 20 clusters of one TSNE/UMAP plot after merging both datasets. Annotations as Cluster 1 to Cluster 10 doesn’t paint a clear picture. It would be good to learn what cell types were captured as sox10:GFP+ and what cell types were captured as foxd3:GFP+. Are they the same cell types? or are they different? If different, how different? I feel like there is a lot of potential in the single-cell data and can reveal a lot of underlying heterogeneity. Expanding on this part can provide a strong foundation for the manuscript.
5. Figure 5C: It appears that the rx3:GFP expression in Krt8 MO injected animals is not lost and is just more disorganized. The number of cells expressing rx3:GFP looks similar between Figure 5B and C. Do the authors think that Krt8 MO does not lead to loss of rx3:GFP expression but those GFP+ cells are just more scattered along the optic cup rim? Are there any changes in the shape and number of rx3+ cells between panels 5B and 5C?
6. Line 420 or Figure 8C: I find it hard to understand how the epithelial, stromal, and endothelial expression of Krt8 and Krt18a.1 were distinguished here? Were markers representative of these three regions used to confirm that the expression was indeed in these 3 cell populations?
7. Quite interesting that both RA signaling inhibition and upregulation seem to elicit the same response in terms of krt8/krt18a.1 expression. Can the authors explain this phenomenon? What is the authors’ hypothesis about what is going on with RA signaling and krt8/krt18a.1?
Minor Comments:
1. Line 250-251: “progressively migrates”. Is there live imaging data showing that krt18a.1+ cells migrate from the EVL to the craniofacial region? Maybe avoid using the word “migrates” without evidence of live imaging microscopy data.
2. Figure 2M, N: For the panels M and N of Figure 2, it may be more appreciated by the community to have the fluorescent color scheme more color-blind friendly. Please refrain from using red and green in the same microscopy image.
3. Line 282-296: Very cool. In the underdeveloped head, were all the head tissues and cell types detected? Did all tissues form but at a lower volume and cell number? Or were certain head cell types lost?
4. Figure 3C: Can you please make the arrowhead a little bigger? It’s a little hard to see.
5. Line 364: Add a citation about the description of DEAB.
6. Line 370: Please add citations here that shows the “well documented teratogenous effects”.
7. Figure 7A–D: In panels A–D, can the authors please make the krt8 staining in red a little brighter? Alternatively, the authors can use a different color scheme as well.
8. Line 426-431: This sentence is too long and cumbersome. Please split this into smaller sentences.
9. Line 431: Is there a figure panel that shows this?
Author Response
Major Comments:
- Line 205-206: Any particular reason why K-Means was used here instead of KNN associated clustering analysis? K-Means will always provide a user-defined number of clusters from scRNAseq data which may not align with biological data. It may be possible that in this manuscript, as shown in Figure 1B, there may be more or less heterogeneity within either of the sox10:EGFP and foxd3:EGFP cells. Maybe try clustering the data using Seurat that uses a shared nearest neighbor approach to pull transcriptionally similar cells together?
Response: Thank you for this comment. We have re-clustered the combined data set using a graph-based approach in the Loupe browser, which does not allow a user-defined number of clusters.
- Line 207-208: “Most of the genes isolated from the Tg(-4.7sox10::EGFP) (52.3%) and 207 Tg(foxd3::GFP) (44.9%) reporter lines were found in Clusters 1 and 2 (FIGURE 1C)” – how were the number of genes quantified here? Was a threshold of expression used to count genes that are expressed above or below a threshold? What would it mean for the other clusters except Clusters 1 and 2 to have lesser number of genes expressed? Is it because there are lesser cells in those clusters or that the data there is not of high quality? Can the authors please explain why having more genes in just 2 clusters is relevant here?
Response: Thank you for this comment. In general, the clusters are numbered by size. Therefore, Cluster 1 will always contain the most cells, and the last cluster will have the fewest cells. The reviewer is correct, and we agree that there should be another stronger and appropriate justification for focusing on clusters of interest. In the revised manuscript, we utilize the expression of specific marker genes as the criteria for which clusters to focus on: “Cell type analysis was based on the upregulated expression of a collection of other POM [periocular mesenchyme] transcription factors (pom+; left panel, FIGURE 1C), namely, pitx2 (paired-like homeodomain 2), foxc1 (forkhead box c1), lmx1b (LIM homeobox transcription factor 1 beta), and eya2 (EYA transcriptional coactivator and phosphatase 2), previously implicated in neural crest cell signaling during early ocular development (reviewed in [Weigele and Bohnsack, 2020]; [Van Der Meulen, et al., 2020]; [Vocking and Famulski, 2022 and 2023])”.
- Line 208-211: The authors show that there are several genes that are differentially expressed between sox10:EGFPand foxd3:EGFP populations in addition to the RA signaling and ASD genes. What about similarities between the two populations? As both label neural crest, it would be interesting to see how similar are these two populations? Can the authors please cluster both datasets together and see if sox10:GFP and foxd3:GFP populations cluster together or do they remain as discrete populations?
Response: Thank you for this comment. Both datasets have been clustered together so that the two neural crest cell subpopulations can be evaluated together.
- In general, it would also be useful to annotate the 10 clusters per dataset or 20 clusters of one TSNE/UMAP plot after merging both datasets. Annotations as Cluster 1 to Cluster 10 doesn’t paint a clear picture. It would be good to learn what cell types were captured as sox10:GFP+ and what cell types were captured as foxd3:GFP+. Are they the same cell types? or are they different? If different, how different? I feel like there is a lot of potential in the single-cell data and can reveal a lot of underlying heterogeneity. Expanding on this part can provide a strong foundation for the manuscript.
Response: Thank you for this comment. Based on our knowledge of the underlying biology of the expected cell types and the genes (or biomarkers, i.e., sox10 and foxd3) used to identify and isolate these subpopulations from the periocular/ocular region of 48-hpf transgenic GFP+ zebrafish, we are highly confident that these cells are neural crest cells. To further investigate their potential heterogeneity, both datasets were clustered together, and clusters of interest were determined by additional periocular mesenchyme (POM) transcription factors previously implicated in neural crest cell signaling during early ocular development and highly expressed in each cluster relative to the other clusters.
- Figure 5C: It appears that the rx3:GFPexpression in Krt8 MO injected animals is not lost and is just more disorganized. The number of cells expressing rx3:GFP looks similar between Figure 5B and C. Do the authors think that Krt8 MO does not lead to loss of rx3:GFP expression but those GFP+ cells are just more scattered along the optic cup rim? Are there any changes in the shape and number of rx3+ cells between panels 5B and 5C?
Response: Thank you for this comment. We believe that the observed reduction in rx3::GFP expression reflects not only the scattering and disorganization of GFP+ neural crest cells along the optic cup rim, as the reviewer correctly implied, but also the notable microphthalmia (small eye size) resulting from the loss of K8 expression due to MO knockdown. In humans, this condition is defined by a small eye with anatomical malformation and markedly shortened axial length, indicating changes in both the shape and number of cells in the underdeveloped eye.
- Line 420 or Figure 8C: I find it hard to understand how the epithelial, stromal, and endothelial expression of Krt8 and Krt18a.1 were distinguished here? Were markers representative of these three regions used to confirm that the expression was indeed in these 3 cell populations?
Response: Thank you for this comment. The distinction of keratin expression in the mature zebrafish cornea was based on morphological examination based on its organization into five layers, with the three most prominent layers (the epithelium, stroma, and endothelium) being easily observed in the adult zebrafish eye by confocal microscopy at 25X magnification. Notably, the establishment of a reliable set of marker genes for developing anterior segment tissues, including the cornea, in zebrafish is largely missing.
- Quite interesting that both RA signaling inhibition and upregulation seem to elicit the same response in terms of krt8/krt18a.1 expression. Can the authors explain this phenomenon? What is the authors’ hypothesis about what is going on with RA signaling and krt8/krt18a.1?
Response: Thank you for this comment. We propose the following rationale for the observed phenomenon and this explanation has been added to the text of the revised manuscript.
“The vitamin A metabolite RA is a potent regulator required for the development of all higher vertebrates from fish to humans. With respect to ocular development, cranial neural crest cells originating from the prosencephalon, diencephalon, and anterior mesencephalon require the tight control of RA levels at multiple steps during development for their precise survival, migration, and eventual differentiation into structures in the midline of the face and anterior segment of the eye. Considering our finding that both increased and decreased RA downregulated krt18a.1 expression, previous studies have shown that increased RA expression (ATRA) decreases cell survival and inhibits ventral cranial neural crest cell migration, while decreased RA expression (DEAB) markedly disrupts both dorsal and ventral cranial neural crest cell migration. In this study, we observed dorsal and ventral patterning of both krt8 and krt18a.1 in the developing zebrafish. Accordingly, following pharmacological insult, the universal decreased expression of krt18a.1 in the periocular region and anterior segment of the eye was reasonably expected. Howbeit, the lack of a significant effect on its canonical binding partner krt8 was not expected.”
“Like that for ocular anterior segment tissues, RA is also a crucial regulator of the development and differentiation of epithelial tissues. However, although K8 and K18 are an essential epithelial cell-specific intermediate filament binding pair, little is known about the influence of RA on K8, while that of K18 has been repeatedly demonstrated [Madame Curie Bioscience Database [Internet]. Austin (TX): Landes Bioscience; 2000-2013. Transcriptional Regulation of Keratin Gene Expression, M Blumenberg; Hans Torma, Dermato-Endocrinology 3:3, 136-140; July/August/September 2011; Ding-Dar Lee, et al., J Cell Physiol. 2009 August ; 220(2): 427–439; James E. Balmer, Rune Blomhoff, Gene expression regulation by retinoic acid, Journal of Lipid Research, Volume 43, Issue 11, 2002, Pages 1773-1808]. Notably, the regulation of individual keratin genes is mediated through the binding of transcription factors to nearby DNA elements [Madame Curie Bioscience Database [Internet]. Austin (TX): Landes Bioscience; 2000-2013. Transcriptional Regulation of Keratin Gene Expression, M Blumenberg; Jiang CK, Epstein HS, Tomic M, Freedberg IM, Blumenberg M. Epithelial-specific keratin gene expression: identification of a 300 base-pair controlling segment. Nucleic Acids Res. 1990 Jan 25;18(2):247–253; MORE]. These transcription factors respond to signaling molecules, such as growth factors, mitogens, hormones, and vitamins, in the extracellular milieu that affect the overall expression of each keratin gene. So, although all known keratins tend to be expressed and act in specific pairs comprising a basic (Type II) and an acidic (Type I) keratin, the corresponding pairwise regulation of their gene expression has not yet been reported. Thus, the differential effect of RA regulation between krt8 and krt18a.1 in the cranial neural crest likely reflects independent regulatory mechanisms that ultimately dictate the expression of each keratin gene independently from the other.”
Minor Comments:
- Line 250-251: “progressively migrates”. Is there live imaging data showing that 1+ cells migrate from the EVL to the craniofacial region? Maybe avoid using the word “migrates” without evidence of live imaging microscopy data.
Response: Thank you for this comment. I agree that the use of this phrase implies an analysis that wasn’t performed here. This phrase has been removed.
- Figure 2M, N: For the panels M and N of Figure 2, it may be more appreciated by the community to have the fluorescent color scheme more color-blind friendly. Please refrain from using red and green in the same microscopy image.
Response: Thank you for this comment. I did not consider this issue. The fluorescent colors in this figure and others with red and green in the same image have been changed to more color-blind friendly schemes.
- Line 282-296: Very cool. In the underdeveloped head, were all the head tissues and cell types detected? Did all tissues form but at a lower volume and cell number? Or were certain head cell types lost?
Response: Thank you for this comment. We did not perform any additional analyses, such as alcian blue or alizarin red staining, to detect head tissues and cell types, an analysis beyond the scope of this ocular neural crest-focused work. Notably, the severely deformed jaw and pharyngeal arch and misshapen underdeveloped head of the mutant fish suggest that these cell types were lost. Indeed, as anophthalmic (missing) eyes were observed with the K18a.1 MO knockdown, it stands to reason that the head tissues were most likely lost.
- Figure 3C: Can you please make the arrowhead a little bigger? It’s a little hard to see.
Response: Thank you for this comment. The arrowhead has been enlarged and color corrected to black for improved visualization.
- Line 364: Add a citation about the description of DEAB.
Response: Thank you for this comment. The citation for DEAB has been added.
- Line 370: Please add citations here that shows the “well documented teratogenous effects”.
Response: Thank you for this comment. Citations for the “well documented teratogenous effects” of RA and DEAB have been added. However, to clarify the intention of this statement, the text has been changed in the revised document to reflect the fact that because of previous studies, these effects were “expected” and not novel findings.
- Figure 7A–D: In panels A–D, can the authors please make the krt8 staining in red a little brighter? Alternatively, the authors can use a different color scheme as well.
Response: The krt8 staining has been brightened as requested, and consistent with Minor Comment #2, the fluorescent color in this figure has been changed to more color-blind friendly scheme.
- Line 426-431: This sentence is too long and cumbersome. Please split this into smaller sentences.
Response: This sentence has been revised to improve clarity as requested.
- Line 431: Is there a figure panel that shows this?
Response: The appropriate figure reference (FIGURE 8M) has been added here.
Reviewer 3 Report
Comments and Suggestions for Authors
In this manuscript, authors used single-cell RNA sequencing analysis to identify differentially expressed genes between sox10-positive craniofacial neural crest cells and foxd3-positive periocular/ocular neural crest cells. They analyzed two candidates, krt8 and krt18a.1, for their expression patterns during normal development and regeneration. They also performed functional analysis, using MO knockdown and RA inhibitors. I think the data is solid, results beautifully presented, conclusions appropriate, and paper well-written.
I have some minor suggestions, mostly about the scRNA seq analysis.
1. Fig 1B is confusing. The two plots used the same color scheme which seems to indicate the two data sets having same clustering. But according to the description in the text, they are from different NCC subpopulations. And indeed, clusters of the same color distributed differently. I recommend combining the two t-SNE plots to one big plot, and label cells of the two data sets with different colors. Ideally there will be non-overlapping foxd3 positive and sox10 positive clusters.
2. Fig 1C, the rationale of pointing out cluster1 and 2 and using them for further analysis are weak. Cluster 1 and cluster 2 are not clear to me. I think top maker analysis is required to assign each cluster to potential cell types, and it will facilitate the later differentially expressed gene analysis. I’m also confused about the phrase “most of the genes isolated” (line 207 and figure legend 1C).
3. Fig 1D, I think pathway enrichment analysis would provide more information and be more convincing.
4. Authors introduced krt8 and krt18a.1 as gene binding pair and then selected them for further analysis. However, they have different expression pattern and indeed as authors demonstrated the two genes have different roles and reaction to RA and wound. Though their different expression patterns and roles are interesting and novel, here I’m not convinced why choosing krt18a.1 over krt18b, which has the same pattern as krt8, making it a better candidate for testing binding pair gene. I also want to know their expression pattern on single cell t-SNE map.
5. The Method section of scRNA seq is too simple. Please provide more information, for example library preparation, quality control, and more details of analysis such as cell number recovered, by-catch cells cleared or not.
6. Please provide sample size n for Fig 3 and Fig 4.
Author Response
- Fig 1B is confusing. The two plots used the same color scheme which seems to indicate the two data sets having same clustering. But according to the description in the text, they are from different NCC subpopulations. And indeed, clusters of the same color distributed differently. I recommend combining the two t-SNE plots to one big plot, and label cells of the two data sets with different colors. Ideally there will be non-overlapping foxd3 positive and sox10 positive clusters.
Response: Thank you for this comment. The two data sets have been combined and we have checked to see if there are non-overlapping foxd3 and sox10 cell populations. The results indicate that these neural crest subpopulations primarily clustered together, with few non-overlapping sox10+ and foxd3+ clusters. This analysis has been included in the revised manuscript and FIGURE 1B has been revised accordingly.
- Fig 1C, the rationale of pointing out cluster1 and 2 and using them for further analysis are weak. Cluster 1 and cluster 2 are not clear to me. I think top maker analysis is required to assign each cluster to potential cell types, and it will facilitate the later differentially expressed gene analysis. I’m also confused about the phrase “most of the genes isolated” (line 207 and figure legend 1C).
Response: Thank you for this comment. In general, the clusters are numbered by size. Therefore, Cluster 1 will always contain the most cells, and the last cluster will have the fewest cells. The reviewer is correct, and we agree that there should be another stronger and appropriate justification for focusing on clusters of interest. In the revised manuscript, we utilize the expression of specific marker genes as the criteria for which clusters to focus on: “Cell type analysis was based on the upregulated expression of a collection of other POM [periocular mesenchyme] transcription factors (pom+; left panel, FIGURE 1C), namely, pitx2 (paired-like homeodomain 2), foxc1 (forkhead box c1), lmx1b (LIM homeobox transcription factor 1 beta), and eya2 (EYA transcriptional coactivator and phosphatase 2), previously implicated in neural crest cell signaling during early ocular development (reviewed in [Weigele and Bohnsack, 2020]; [Van Der Meulen, et al., 2020]; [Vocking and Famulski, 2022 and 2023])”.
- Fig 1D, I think pathway enrichment analysis would provide more information and be more convincing.
Response: Thank you for this comment. We performed pathway enrichment analyses using the online tool Metascape [(Zhou et al. Nature Commun. 2019 10(1):1523)]. This analysis has been included in the revised manuscript and FIGURE 1D has been revised accordingly.
- Authors introduced krt8and 1 as gene binding pair and then selected them for further analysis. However, they have different expression pattern and indeed as authors demonstrated the two genes have different roles and reaction to RA and wound. Though their different expression patterns and roles are interesting and novel, here I’m not convinced why choosing krt18a.1over krt18b, which has the same pattern as krt8, making it a better candidate for testing binding pair gene. I also want to know their expression pattern on single cell t-SNE map.
Response: Thank you for this comment. We agree that an analysis of krt18b in the ocular neural crest is reasonable. However, the decision to explore krt18a.1 over krt18 was primarily based on reference to the Zebrafish Information Network (ZFIN), a database of reported and predicted genetic and genomic information for the model organism zebrafish (Danio rerio), which showed that krt8 and krt18a.1 expression had previously been detected in the eyes of adult zebrafish at >90 days postfertilization (dpf) [Garcia, D.M., Bauer, H., Dietz, T., Schubert, T., Markl, J., and Schaffeld, M., 2005; Vihtelic, T.S., Fadool, J.M., Gao, J., Thornton, K.A., Hyde, D.R., and Wistow, G., 2005], whereas krt18b expression was previously detected throughout the entire juvenile zebrafish at 30-45 dpf but not in the eyes. Additionally, both krt18a.1 and 18b are human KRT18 orthologs and transcript variants of the same protein equally upregulated in the POM neural crest during ocular development (FIGURE 1F), so we only examined krt8 and krt18a.1 as representatives of the obligate binding pair and most widely dispersed members of the intermediate filament gene family krt8 and krt18 to obtain a general understanding of their roles in ocular neural crest development and anterior segment formation. This explanation has been added to the text of the revised manuscript.
- The Method section of scRNA seq is too simple. Please provide more information, for example library preparation, quality control, and more details of analysis such as cell number recovered, by-catch cells cleared or not.
Response: Thank you for this comment. We have consulted with the Northwestern University Sequencing Core (NUSeq), and the protocol for conducting the transcriptomics analysis has been rewritten in more detail as requested.
- Please provide sample size n for Fig 3 and Fig 4.
Response: Thank you for this comment. The numbers of fish (embryos and adults) used in this study are indicated in the Methods section as follows:
- Wholemount in situ hybridization and immunostaining:
Line 157- “at 50 embryos/tube per experimental condition”
- Pharmacological treatments:
Line 214 – “Casper embryos, at 50 to 100 embryos per treatment group”
- Morpholino oligonucleotide injections: (ref: FIGURE 3 and FIGURE 4)
Line 237 – “Each experiment was conducted a minimum of 3 times with at least 25 to 50 embryos per group”
- Corneal abrasion and immunohistochemistry:
Line 247 – “Each experiment was repeated 3-4 times, harvesting 4-6 fish at each collection period”
Round 2
Reviewer 1 Report
Comments and Suggestions for Authors
The corrected version of the article may be recommended for publication in the journal Cells. The authors took note of most of the comments and made appropriate corrections. However, authors should highlight the corrections made in color to speed up the process of re-reviewing the work. Unfortunately, the authors did not indicate how many animals were used in this work. The following corrections were made in the revised version of the article:
1. Unfortunately, the authors did not indicate how many animals were used in this work;
2. The authors detailed the method of transcriptome analysis and added a sequencing protocol, which significantly complemented the methodological part of the work.
3. The authors included the current protocol of in situ hybridization in the Materials and methods sections, which significantly improved the content of the methodological part of the work.
4. Clarifications have been made to the algorithm of calorimetric estimation with a demonstration of this approach in the Results section.
5. The authors presented photographs in Figs. 8 and 9 with a more specific IHC labelling, but the authors did not follow the recommendations regarding the control of antibody specificity.
6. The authors have made appropriate changes to the Materials and methods section with reference to the results of the study.
7. Unfortunately, the authors did not provide data on the specificity of these antibodies for Danio, in particular, Western immunoblotting with protein loading control and/or positive control.
8. Stylistic correction of the language throughout the article was carried out
9. In the Discussion section, the authors, in accordance with the recommendations, have identified subsections that are recommended to be numbered in accordance with the general content of the article, for example, highlight sections 3.1, 3.2, 3.3, etc.
Comments on the Quality of English Language
English language editing is recommended
Reviewer 2 Report
Comments and Suggestions for Authors
Well organized and a much improved version of the manuscript. The authors took into account all of my comments and suggestions and have used those as a guide to considerably improve the manuscript. The manuscript now does a lot of justice to the single-cell RNAseq analysis and revealed a lot of previously unknown mechanisms underlying neural crest migration. In it's current form, I am satisfied and wholeheartedly recommend this manuscript for publication. Congratulations!
Author Response
Thank you. We really appreciate your helpful comments on the first round of review, which helped to vastly improve this submission.
Reviewer 3 Report
Comments and Suggestions for Authors
Questions are answered. I agree to publish it.
Author Response
Thank you. Your helpful comments on the first round of review helped to vastly improve this submission.